# Optimal Decision-Making Based on Prediction Sets

Tao Wang [1]  Edgar Dobriban [1]

## Abstract

Prediction sets can wrap around any ML model to cover unknown test outcomes with a guaranteed probability. Yet, it remains unclear how to use them optimally for downstream decision-making. Here, we propose a decision-theoretic framework that seeks to minimize the expected loss (risk) against a worst-case distribution consistent with the prediction set's coverage guarantee. We first characterize the minimax optimal policy for a fixed prediction set, showing that it balances the worst-case loss inside the set with a penalty for potential losses outside the set. Building on this, we derive the optimal prediction set construction that minimizes the resulting robust risk subject to a coverage constraint. Finally, we introduce Risk-Optimal Conformal Prediction (ROCP), a practical algorithm that targets these risk-minimizing sets while maintaining finite-sample distribution-free marginal coverage. Empirical evaluations on medical diagnosis and a toy static hazard-decision benchmark demonstrate that ROCP reduces critical mistakes compared to baselines, particularly when out-of-set errors are costly. The source code to reproduce our experiments is available at https://github.com/TaoWangPenn/Risk-Optimal-Conformal-Prediction.

## 1. Introduction

Predictions are rarely an end in themselves: they are used to select actions. In medicine, a model's diagnosis informs treatment; in robotics, perception informs control. In such settings, the downstream objective is not "accuracy" per se but low *decision loss*. The classical statistical decision-theoretic prescription is clear: if the conditional law of

the outcomes $Y$ given the features $x$, $P(Y \mid X = x)$, were known, one would choose the Bayes action minimizing expected loss $\mathbb{E}[\ell(a, Y) \mid X = x]$ (Wald, 1945; 1949; Lehmann & Casella, 1998). In modern ML pipelines, however, the conditional law is unknown and predictive models are imperfect. This has motivated a great deal of work on optimal decision making under uncertainty, see e.g., Keith & Ahner (2021); Elmachtoub & Grigas (2022); Zhao et al. (2021), etc. However, many of these works still assume that one can obtain "correct" probability predictions in some weak sense, e.g., consistent or calibrated predictions.

To handle the case that the data distribution is completely unknown, the area of distribution-free uncertainty quantification and conformal prediction has emerged (e.g., Wilks, 1941; Wald, 1943; Vovk et al., 1999; 2005; Lei et al., 2013; Angelopoulos et al., 2023, etc). Given i.i.d. data (and even under the weaker assumption of exchangeability), conformal methods output prediction sets $C(x) \subseteq \mathcal{Y}$ satisfying a finite-sample, distribution-free marginal guarantee $\Pr\{Y \in C(X)\} \geq 1 - \alpha$ (Vovk et al., 2005; Angelopoulos et al., 2023). Yet coverage by itself does not specify how to act. This motivates a central question at the prediction–action interface: how should one make *provably safe and effective decisions* when the only reliable information about $Y$ comes from a prediction set with a coverage guarantee?

Recent work by Kiyani et al. (2025) suggests choosing a *max–min* optimal action, minimizing the worst-case loss over $y \in C(x)$. They show that this rule is optimal for *quantile-style* objectives, where the agent only cares about performance on a $1 - \alpha$ fraction of outcomes (Kiyani et al., 2025). However, the more standard notion of expected loss is sensitive to rare but catastrophic events: even if $Y \notin C(x)$ occurs with probability at most $\alpha$, the loss incurred on that event may be orders of magnitude larger than any in-set loss. In such regimes, a purely in-set max–min rule can be brittle because it has no incentive to hedge against the $\alpha$ mass that is *not* constrained by the set.

**From prediction sets to minimax-optimal actions.** This paper develops a decision-theoretic framework that makes this trade-off explicit. Our starting point is a two-player game between the decision maker and nature. For a fixed set $S \subseteq \mathcal{Y}$ and action $a \in \mathcal{A}$, nature may choose any distribution on $\mathcal{Y}$ that places at least $1 - \alpha$ probability

---

[1]Department of Statistics and Data Science, Wharton School, University of Pennsylvania. Correspondence to: Tao Wang <tawan@wharton.upenn.edu>, Edgar Dobriban <dobriban@wharton.upenn.edu>.

on $S$. The resulting worst-case expected loss is the function $L_S(a; \alpha)$ from (2). We show that this worst-case expectation admits a simple closed form (Lemma 2.1): $L_S(a; \alpha) = \ell_S^{\text{in}}(a) + \alpha(\ell_S^{\text{out}}(a) - \ell_S^{\text{in}}(a))_+$; where $\ell^{\text{in}}$ and $\ell^{\text{out}}$ are the maximum losses inside and outside of $S$, respectively. This expression has a transparent interpretation. The dominant term is the worst loss *inside $S$* (recovering the rule from (Kiyani et al., 2025)); but if the worst loss *outside $S$* is larger, the decision maker must pay an additional $\alpha$-weighted penalty. Thus, unlike a pure max–min rule, the optimal action hedges against catastrophic out-of-set outcomes whenever they can materially affect expected risk.

Lifting this pointwise characterization to the original prediction-set pipeline, we derive the minimax-optimal policy $\pi^\star$ for any set-valued predictor $C$ (Theorem 2.2). This yields decision rule that reduces to the in-set max–min rule when $\alpha = 0$, but differs sharply in high-stakes regimes where out-of-set mistakes are costly.

**Designing prediction sets for decision quality.** The second half of the paper addresses the natural next question: if prediction sets will be used to drive decisions, how should we choose them? Decision-agnostic conformal sets are typically optimized for surrogate criteria such as size or top-$k$ mass (see e.g, Sadinle et al., 2019; Romano et al., 2020; Wang et al., 2025, etc), but these objectives need not align with downstream loss. Motivated by our minimax characterization, we formulate a population optimization problem that directly minimizes the robust risk induced by the optimal downstream decision rule, subject to a coverage constraint (6); similarly to the one for the different quantile-based objective in Kiyani et al. (2025). Using duality theory for integral functionals, we characterize the optimal coverage assignment (Theorem 3.3).

**A finite-sample algorithm with distribution-free coverage.** To translate the oracle characterization into a practical procedure, we introduce *Risk-Optimal Conformal Prediction (ROCP)*. ROCP uses any black-box probabilistic model to construct estimates of the population quantities in our oracle decision rule, and then uses a held-out calibration set to ensure coverage via conformal prediction (Algorithm 1). This guarantees finite-sample marginal coverage under exchangeability (Theorem 4.1), while targeting the decision-theoretic optimum as the model improves. Empirically, ROCP consistently reduces worst-case risks and critical mistake rates relative to risk-averse max–min baselines and best-response approaches, with the largest gains arising precisely in regimes where the $\alpha$ fraction of out-of-set mass can induce catastrophic losses.

Together, these ideas provide an approach to connect prediction sets and actions, as illustrated in Figure 3.

## 1.1. Related Work

There is a great deal of related work. Due to space limitations, we discuss some of it in Appendix B.

**Decision making under set-valued uncertainty.** A natural way to act given a set $C(x)$ is to choose an action that is robust to all $y \in C(x)$, leading to max–min decision rules. This principle is central in robust optimization, where uncertainty sets replace probabilistic models (Chan & Kaw, 2020; Chan et al., 2023; 2024; Patel et al., 2024; Johnstone & Cox, 2021; Yeh et al., 2024). Our formulation is closely related but distinct: we do not assume $Y \in C(x)$ surely. Instead, the uncertainty comes with a *coverage constraint* that leaves an $\alpha$ fraction of probability mass unconstrained. The resulting optimal rule is therefore robust only *in expectation* over the worst-case distribution consistent with coverage, which produces the additional out-of-set penalty term in Lemma 2.1. This distinction can matter in applications where rare out-of-set errors have disproportionate cost.

Conformalization ideas have been used by Chen et al. (2026) to study conformal uncertainty sets for robust optimization, learning polyhedral prediction regions whose geometry is adapted to the downstream robust loss. Complementarily, Zhou & Zhu (2025) address the question of how to choose the robustness level itself in robust predict-then-optimize pipelines. Zhou et al. (2025) introduce CREDO, a conformalized framework for assessing the risk of a prescribed candidate decision.

**Risk-averse objectives and our companion work.** Prior work by Kiyani et al. (2025) studies the prediction–action interface for risk-averse agents who optimize a quantile-style objective (value-at-risk) and proves that, under a marginal coverage constraint, the max–min policy is minimax-optimal for that criterion. It also derives population-optimal prediction sets and a finite-sample algorithm (RAC) tailored to this quantile objective. Our paper can be viewed as the expectation-risk counterpart: we replace the quantile objective by expected loss, which fundamentally changes the minimax structure. In particular, expected loss forces the decision maker to account for the $\alpha$ mass outside the set whenever it can generate larger losses, leading to a different optimal policy and a different optimal set construction.

Although the set-design problem in both this paper and Kiyani et al. (2025) admit a one-dimensional dual parameter, the duality arguments are technically different. Kiyani et al. (2025) reparametrize the coverage assignment via an auxiliary step function reducing the problem to an infinite-dimensional linear program which by LP duality theory yields the threshold form solution. In contrast, our objective is an integral functional. To justify strong duality and the interchange of infimum and expectation, we work

in the normal-integrand framework and invoke Fenchel-Rockafellar duality through a randomized-kernel relaxation, followed by derandomization to recover a deterministic optimizer. Empirically, this difference is most pronounced in high-stakes regimes with highly asymmetric losses, where small miscoverage probabilities can still dominate the expected risk.

## 2. Optimal action selection from prediction sets

### 2.1. Problem Formulation

In this section, we formalize the goal of minimizing the expected loss, leveraging a prediction set. Let $\mathcal{X}$ and $\mathcal{Y}$ be the feature and outcome spaces, and let $\mathcal{A}$ be the action space. For instance, the actions could be the outcomes: $\mathcal{A} = \mathcal{Y}$.

A prediction set is a map $C : \mathcal{X} \to 2^{\mathcal{Y}}$, assigning to each $x \in \mathcal{X}$ a set $C(x) \subseteq \mathcal{Y}$. A *policy* $\pi : 2^{\mathcal{Y}} \times [0,1] \to \mathcal{A}$ takes as input a set $S \subseteq \mathcal{Y}$ together with a miscoverage parameter $\alpha' \in [0,1]$ (equivalently, coverage $t = 1 - \alpha'$), and outputs an action $\pi(S, \alpha') \in \mathcal{A}$. When a single global miscoverage level $\alpha$ is fixed, we write $\pi(S) := \pi(S, \alpha)$ for brevity. Let $\Pi$ denote the class of such policies. The loss of the decision maker depends on both the chosen action $a$ and the label $y$, and is captured by a fixed loss function $\ell : \mathcal{A} \times \mathcal{Y} \to \mathbb{R}_+ := [0, \infty)$.

We assume that the decision maker has access to a prediction set $C(x)$ that is guaranteed to contain the true label with high probability. Specifically, in order to derive our oracle-optimal decision policies, we will consider an idealized prediction set that satisfies a conditional coverage property $\mathbb{P}_{(X,Y)\sim P}\big(Y \in C(X) \mid X = x\big) \geq 1 - \alpha$, $P_X$ a.e. $x$ for some $\alpha \in (0,1)$. In practice, such conditional coverage properties are not generally possible for continuous feature spaces, but rather only possible for discrete feature spaces or approximately in continuous domains, see e.g., Vovk (2012); Lei & Wasserman (2014); Barber et al. (2020); Guan (2023); Gibbs et al. (2025); Joshi et al. (2025a). However, we emphasize that the idealization will only be considered for technical reasons in order to derive a clean form for the oracle optimal decision policies. In practice, we will use those policies along with prediction sets satisfying the standard marginal coverage conditions $\mathbb{P}_{(X,Y)\sim P}\big(Y \in C(X)\big) \geq 1 - \alpha$ (Vovk et al., 2005; Angelopoulos et al., 2023). We will see that, despite motivating the oracle from a stronger condition, it will empirically lead to better decisions even under the standard realistic conditions. We typically consider $\alpha$ to be small, such as 0.05.

Given a prediction set, a canonical interpretation is that the outcomes in the prediction set are all plausible. However, how should one use these outcomes in a downstream decision-making task? Our goal is precisely to address this task, in the standard framework of statistical decision theory, which focuses on the risk, namely the expected loss (see e.g., Wald, 1945; 1949; Lehmann & Casella, 1998, etc.). Our first goal is to characterize optimal decision policies subject to choosing actions from prediction sets with a fixed coverage level.

Fix $\alpha \in [0,1)$, and define $\mathcal{P}_\alpha := \mathcal{P}_\alpha(C)$ as the set of all the data distributions that are consistent with conditional coverage[1]

$$\mathcal{P}_\alpha := \Big\{ P \text{ on } \mathcal{X} \times \mathcal{Y} : P\big(Y \in C(X) \mid X = x\big) \geq 1 - \alpha, \text{ a.e. } x\Big\}.$$

As mentioned above, to derive our oracle-optimal rule, we consider an idealized setting where the true distribution $P$ is unknown but it belongs to $\mathcal{P}_\alpha$. We aim to find the optimal policy minimizing the risk when choosing actions from the prediction set $C$, in the worst case over all such distributions. This leads to the problem of solving, over all policies $\pi : 2^{\mathcal{Y}} \to \mathcal{A}$:

$$\inf_{\pi} \sup_{P \in \mathcal{P}_\alpha} \mathbb{E}_{(X,Y)\sim P}\Big[\ell\big(\pi(C(X)), Y\big)\Big]. \tag{1}$$

This can be viewed as a two-person game, played between the analyst, who wishes to choose the best policy $\pi$, and "nature", which can set the data distribution $P$ to be adversarial. Kiyani et al. (2025) study an analogous problem, minimizing an expected quantile of the loss under a marginal coverage constraint. In contrast, we study minimizing the expectation of the loss; which is motivated by the standard formulation of statistical decision theory (see e.g., Wald, 1945; 1949; Lehmann & Casella, 1998, etc.). Our solution turns out to be different in intriguing ways.

### 2.2. Optimal Policies

For a set $S \subseteq \mathcal{Y}$ and $a \in \mathcal{A}$, define $\ell_S^{\text{in}}(a) := \sup_{y \in S} \ell(a, y)$, to be the maximum[2] of the loss achieved by action $a$ over the set $S$. Similarly, define $\ell_S^{\text{out}}(a) := \sup_{y \notin S} \ell(a, y)$ to be the maximum achieved outside of the set $S$, with the convention that $\ell_{\mathcal{Y}}^{\text{out}}(a) := \ell_{\mathcal{Y}}^{\text{in}}(a)$. For an action $a$ and a set $S \subseteq \mathcal{Y}$, define the worst-case expected loss—over probability distributions $Q$ for which the set $S$ achieves coverage—under a miscoverage level of $\alpha$ by

$$L_S(a; \alpha) = \sup_{Q(\cdot):\, Q(S) \geq 1-\alpha} \mathbb{E}_{Y \sim Q}[\ell(a, Y)]. \tag{2}$$

We will only apply $L_S(a; \alpha)$ to sets $S$ for which the constraint set $\{Q : Q(S) \geq 1 - \alpha\}$ is nonempty (in particular,

---

[1]From now on, a.e. $x$ always refers to $P_X$ a.e. $x$ for brevity.

[2]If $\mathcal{Y}$ is finite, the suprema in $\ell_S^{\text{in}}$ and $\ell_S^{\text{out}}$ are attained. More generally, attainment holds if $\ell(a, \cdot)$ is upper semicontinuous and the optimization domain is compact (e.g., if $S$ and $S^c$ are compact subsets of a compact $\mathcal{Y}$). Throughout we assume the suprema are finite, and when we require attainment we state it explicitly.

$S \neq \varnothing$ whenever $\alpha < 1$).[3]

This notion will turn out to be an important intermediate quantity in our analysis, because it characterizes the worst-case expected loss that can be induced by a probability distribution subject to coverage; for a fixed predicted set $S \subset \mathcal{Y}$. We will use this repeatedly in our development. Therefore, it will be important to find a simpler expression for it. Fortunately, it turns out that this is possible, as shown by the following result.

**Lemma 2.1** (Worst-case expected loss under miscoverage level of $\alpha$). *For any $S \subseteq \mathcal{Y}$ and any $a \in \mathcal{A}$,*

$$L_S(a; \alpha) = \ell_S^{\text{in}}(a) + \alpha \left( \ell_S^{\text{out}}(a) - \ell_S^{\text{in}}(a) \right)_+ \quad (3)$$

*where $(t)_+ = \max\{t, 0\}$.*

The formula in (3) has an insightful interpretation: the worst-case loss first looks at the worst-case outcome inside the prediction set $S$ (i.e., $\ell_S^{\text{in}}(a)$), which represents the worst-case loss over the more likely set with probability at least $1 - \alpha$. This is then compared with the worst-case outcome outside of the prediction set $S$—through $\ell_S^{\text{out}}(a)$—and if the loss outside is larger, then a penalty of $\left( \ell_S^{\text{out}}(a) - \ell_S^{\text{in}}(a) \right)_+$—multiplied by the typically small $\alpha$ is added. In other words, the worst-case loss is mainly determined by the losses within the prediction set, but it is also penalized (by a small amount) by the losses outside of the prediction set. In contrast, the solution to the analogous problem for the quantile from Kiyani et al. (2025) amounts to only the first component, namely $\ell_S^{\text{in}}(a)$. The proof (with all proofs) is deferred to the Appendix C.

Equipped with this result, we can now characterize[4] the optimal policy and the worst-case distribution for our original problem from (1):

**Theorem 2.2** (Optimal policy and risk). *If for every $x \in \mathcal{X}$, the function $a \mapsto L_{C(x)}(a; \alpha)$ attains its minimum [5], then optimal policies $\pi^\star$ to the problem (1) have the form [6]*

$$\pi^\star(C(x)) \in \arg\min_{a \in \mathcal{A}} L_{C(x)}(a; \alpha), \ x \in \mathcal{X}, \quad (4)$$

*and the minimax risk is*

$$\inf_\pi \sup_{P \in \mathcal{P}_\alpha} \mathbb{E} \, \ell\big(\pi(C(X)), Y\big) = \sup_{x \in \mathcal{X}} \min_{a \in \mathcal{A}} L_{C(x)}(a; \alpha). \quad (5)$$

---

[3]When $S = \emptyset$ and $\alpha < 1$, there is no probability distribution satisfying the constraint. Therefore, we can define $L_S(a; \alpha)$ as $\infty$.

[4]We tacitly restrict to measurable set-valued predictors $C$ and policies $\pi$ for which $\ell(\pi(C(X)), Y)$ is measurable; see Appendix C for formal conventions.

[5]If $\mathcal{A}$ is compact and $a \mapsto \ell(a, y)$ is lower semicontinuous for each $y$, then the function $a \mapsto L_S(a; \alpha)$ attains its minimum.

[6]Existence of measurable selection is discussed in Appendix C; see Remark C.1 after the proof of Theorem 2.2.

*Moreover, if in addition the suprema in $\ell_S^{\text{in}}$ and $\ell_S^{\text{out}}$ are attained for all $a$, and the outer supremum over $x$ in (5) is attained, then a worst-case $P^\star \in \mathcal{P}_\alpha$ can be chosen as follows. Let $P_X = \delta_{x^\star}$ for some $x^\star \in \arg\max_{x \in \mathcal{X}} \min_{a \in \mathcal{A}} L_{C(x)}(a; \alpha)$ and, writing $S^\star = C(x^\star)$,*

$$Y|X = x^\star \sim \begin{cases} (1 - \alpha)\delta_{y_i} + \alpha\delta_{y_o}, \text{if } S^\star \neq \mathcal{Y} \ \& \ \ell_\star^{\text{out}} > \ell_\star^{\text{in}}, \\ \delta_{y_i}, \text{otherwise}, \end{cases}$$

*where $\ell_\star^{\text{in}} = \ell_{S^\star}^{\text{in}}(\pi^\star(S^\star))$ and $\ell_\star^{\text{out}} = \ell_{S^\star}^{\text{out}}(\pi^\star(S^\star))$, and $y_i \in \arg\max_{y \in S^\star} \ell(\pi^\star(S^\star), y)$ while (in the first case) $y_o \in \arg\max_{y \notin S^\star} \ell(\pi^\star(S^\star), y)$.*

Theorem 2.2 states that the adversary will always concentrate all the feature mass at a single point $x^\star$, and the resulting risk reduces to $\inf_{a \in \mathcal{A}} L_{C(x^\star)}(a; \alpha)$. Hence, when the decision maker wants to make the decision based on prediction sets $C(x)$, $x \in \mathcal{X}$ that contain the actual label with high probability $1 - \alpha$, the minimax optimal policy and the corresponding per-$x$ robust risk are

$$a^\star(x) := \pi^\star(C(x)), \quad R(C(x), \alpha) := \min_{a \in \mathcal{A}} L_{C(x)}(a; \alpha).$$

*Remark* 2.3 (Conditional versus marginal coverage). Problem (1) uses an idealized conditional-coverage oracle because it yields a clean pointwise game and the closed-form local robust loss in Lemma 2.1 and Theorem 2.2. Under the marginal class

$$\mathcal{P}_\alpha^{\text{m}}(C) := \{P \text{ on } \mathcal{X} \times \mathcal{Y} : \Pr(Y \in C(X)) \geq 1 - \alpha\},$$

the exact robust action game is global: it couples covariates through the worst in-set and out-of-set losses and no longer decomposes pointwise in $x$. Hence the local robust loss $L_{C(x)}(a; \alpha)$ is not the exact marginal oracle, and a strict gap can occur. In Appendix C, we derive the marginal objective and show that, for any fixed prediction set $C$, the policy from Theorem 2.2 is a $(2 - \alpha)$-approximation to the optimal marginal policy.

Now we turn to the question of designing prediction sets that are suitable for decision-making. First, motivated by the above characterization, we consider a hypothetical setting where the true distribution $P$ was known. We will later show how to apply this idea when $P$ is unknown. As we already mentioned, the oracle optimal decision-theoretic characterization is phrased in terms of *conditional* miscoverage, which is generally impossible (Vovk, 2012; Foygel Barber et al., 2021). In practice, we will apply the method to a prediction set with a standard *marginal* coverage guarantee $\mathbb{P}_P(Y \in C(X)) \geq 1 - \alpha$; and we will argue empirically that the performance improves.

In the oracle set-design problem below (where $P$ is known), we allow the decision rule to depend on a *coverage assignment* $t : \mathcal{X} \to [0, 1]$, interpreted as the conditional

coverage level that is certified at covariate value $x$. Concretely, we require that $\Pr\{Y \in C(X) \mid X = x\} \geq t(x)$ for $P_X$-a.e. $x$. By the tower property, this implies the marginal coverage bound $\Pr\{Y \in C(X)\} \geq \mathbb{E}[t(X)]$. Given such a certified level $t(x)$, the corresponding local miscoverage budget is $1 - t(x)$ and the robust risk at $x$ is $R(C(x), 1 - t(x)) = \min_{a \in \mathcal{A}} L_{C(x)}(a; 1 - t(x))$. We therefore consider the oracle set-design problem

$$\min_{C(\cdot),\, t(\cdot)} \quad \mathbb{E}\big[R\big(C(X), 1 - t(X)\big)\big]$$
$$\text{s.\,t. } \Pr\{Y \in C(X) \mid X = x\} \geq t(x) \text{ for } P_X\text{-a.e. } x, \quad (6)$$
$$\mathbb{E}[t(X)] \geq 1 - \alpha.$$

Next, we will explain how to solve this problem.

## 3. The Optimal Prediction Sets

In this section, we characterize the optimal prediction sets for the problem (6) by first studying the pointwise-across-$x$ problem and introducing a reparametrization (see (9)) which we then solve using techniques from duality theory.

Starting from this section, we drop the subscript $P$ on $\mathbb{P}_P$ and $\mathbb{E}_P$ for simplicity. Our analysis begins with a *pointwise* problem: fix a feature value $x \in \mathcal{X}$ and a target *certified* conditional coverage level $t \in (0, 1]$. We will design a set $C(x) \subseteq \mathcal{Y}$ subject to the constraint $\Pr\{Y \in C(x) \mid X = x\} \geq t$, and evaluate decisions under the corresponding miscoverage budget $1 - t$ via the robust risk $R(C(x), 1 - t) = \min_{a \in \mathcal{A}} L_{C(x)}(a; 1 - t)$.

For $a \in \mathcal{A}$, define the maximal loss that action $a$ can incur as $M(a) := \sup_{y \in \mathcal{Y}} \ell(a, y) < \infty$, and define the conditional $t$-quantile of the loss at $(x, a)$ as

$$Q_t^x(a) := \inf\big\{\theta \in \mathbb{R} : \mathbb{P}(\ell(a, Y) \leq \theta \mid X = x) \geq t\big\}.$$

Also, consider the loss sublevel set for some $\theta$, namely $S_\theta(a) := \{y \in \mathcal{Y} : \ell(a, y) \leq \theta\}$. Intuitively, $S_{Q_t^x(a)}(a)$ are the lowest-loss labels under action $a$, with $x$-conditional coverage of at least $t$. This is the natural candidate feasible set with small worst-case in-set loss $\sup_{y \in C(x)} \ell(a, y)$. Moreover, under the constraint $\mathbb{P}(Y \in C(x) \mid X = x) \geq t$, an adversary in the definition of $L_{C(x)}(a; 1 - t)$ may place probability $t$ on a worst in-set label (with loss $Q_t^x(a)$ when $C(x) = S_{Q_t^x(a)}(a)$) and the remaining probability $1 - t$ on a label achieving the maximal loss $M(a)$, leading to the pointwise objective $t Q_t^x(a) + (1 - t) M(a)$. Then the corresponding pointwise optimal action is

$$a(x, t) \in \arg\min_{a \in \mathcal{A}} \{t Q_t^x(a) + (1 - t) M(a)\}. \quad (7)$$

The associated threshold below which the loss values are included is $\theta(x, t) := Q_t^x(a(x, t))$; and the resulting optimal set $C(x, t)$ is

$$S_{\theta(x,t)}\big(a(x, t)\big) = \{y \in \mathcal{Y} : \ell\big(a(x, t), y\big) \leq \theta(x, t)\}. \quad (8)$$

The following proposition summarizes this formally. It is an analogue of Proposition 3.1 in (Kiyani et al., 2025), with the difference that our formulation for expected risk necessitates explicitly accounting for the $(1 - t)$ probability mass falling outside the set, yielding the additional $(1 - t)M(a)$ term absent from their formulation.

**Proposition 3.1.** *Fix an instance $x \in \mathcal{X}$ and a coverage value $t \in (0, 1]$. Assume the minimum in (7) exists,[7] then among all the sets $C \subseteq \mathcal{Y}$ that have coverage at least $t$, i.e. $\Pr(Y \in C \mid X = x) \geq t$, the set $C(x, t)$ from (8) achieves the smallest risk $R(C, 1 - t)$ with*

$$R(C(x, t), 1 - t) = t\, \theta(x, t) + (1 - t)\, M\big(a(x, t)\big).$$

*Further, if the suprema defining $\ell_{C(x,t)}^{\mathrm{in}}(a(x, t))$ and $M(a(x, t))$ are attained, then the worst–case conditional law achieving the inner supremum in $L_{C(x,t)}(a(x, t); 1 - t)$ can be taken to place probability $t$ on a point $y_{\mathrm{i}} \in \arg\max_{y \in C(x,t)} \ell(a(x, t), y)$ and probability $1 - t$ on a point $y_{\mathrm{o}} \in \arg\max_{y \in \mathcal{Y}} \ell(a(x, t), y)$.*

*Remark* 3.2 (Edge cases). At $t = 0$ the constraint is vacuous, i.e., every measurable $C \subseteq \mathcal{Y}$ is feasible, and the value reduces to $\min_{a \in \mathcal{A}} M(a)$, independent of $C$. For later use, when $t = 0$ we fix any minimizer $a(x, 0) \in \arg\min_{a \in \mathcal{A}} M(a)$ and set $\theta(x, 0) := M(a(x, 0))$ and $C(x, 0) := \mathcal{Y}$.

Proposition 3.1 allows us to reparametrize (6) in terms of a pointwise coverage assignment $t : \mathcal{X} \to [0, 1]$. Formally, (6) has the following equivalent reformulation:

$$\mathrm{VAL}(\alpha) := \inf_{\substack{t:\mathcal{X}\to[0,1] \text{ measurable} \\ \mathbb{E}[t(X)] \geq 1-\alpha}} \mathbb{E}\big[V_X\big(t(X)\big)\big]. \quad (9)$$

where we define

$$V_x(t) := \min_{a \in \mathcal{A}} \big\{t\, Q_t^x(a) + (1 - t)\, M(a)\big\}, \quad t \in (0, 1], \quad (10)$$

and set $V_x(0) := \min_{a \in \mathcal{A}} M(a)$ as in Remark 3.2. Letting $t^\star$ be the optimum, the optimal actions are $a^\star(x) = a(x, t^\star(x))$, and the optimal prediction set is:

$$C^\star(x) = C\big(x, t^\star(x)\big) \quad (11)$$
$$= \Big\{y \in \mathcal{Y} : \ell\big(a(x, t^\star(x)), y\big) \leq \theta\big(x, t^\star(x)\big)\Big\}.$$

To solve this problem, we adopt a duality-based approach in the spirit of Kiyani et al. (2025). However, their proof strategy does not transfer directly to our setting. They reduce their problem to an infinite-dimensional linear program and leverage LP duality theory, which does not directly apply here. Instead, we introduce a different technical approach: we work in the normal-integrand framework and invoke

---

[7]This holds, e.g., if $\mathcal{A}$ is compact and $a \mapsto Q_t^x(a)$ and $a \mapsto M(a)$ are lower semicontinuous.

Fenchel–Rockafellar duality through a randomized-kernel relaxation, followed by derandomization to recover a deterministic optimizer. The resulting solution still has a simple one-dimensional structure: it is indexed by a scalar parameter $\beta \in \mathbb{R}_+$ and a selector function $g : \mathcal{X} \times \mathbb{R}_+ \to [0,1]$ defined as[8]

$$g(x,\beta) \ = \ \arg \min_{t \in [0,1]} \big\{ V_x(t) \ - \ \beta\,t \big\}, \qquad (12)$$

**Theorem 3.3.** *Fix $\alpha \in (0,1)$. Assume $P_X$ is non-atomic, that $(x,t) \mapsto V_x(t)$ in (10) is a normal integrand,[9] and that, for each $x$, the minimum in the definition of $V_x(t)$ exists for every $t \in [0,1]$. Then there exists $\beta^\star \geq 0$ such that*

$$t^\star(x) \ := \ g(x,\beta^\star)$$

*solves the population problem (9). Consequently, the optimal prediction sets are obtained using $t^\star(x)$ from (11), with actions $a^\star(x) = a\big(x, t^\star(x)\big)$ as in (7).*

*Moreover, $\beta^\star$ is characterized by the scalar equation*

$$\mathbb{E}_X\big[g(X,\beta)\big] \ = \ 1 - \alpha, \qquad (13)$$

*in the sense that any optimal $\beta^\star > 0$ must satisfy (13), and conversely, any $\beta \geq 0$ satisfying (13) yields an optimal $t^\star(x) = g(x,\beta)$. If (13) has no solution with $\beta > 0$ but $\mathbb{E}[g(X,0)] \geq 1 - \alpha$, then $\beta^\star = 0$.*

## 4. Risk–optimal conformal prediction

Theorem 3.3 characterizes an oracle optimal prediction set, assuming the true distribution is known. This section constructs a conformal prediction set that emulates this oracle based on data. While the approximation guarantee to the optimum is, in general, challenging to establish, the conformal prediction set provides a finite-sample guarantee–namely, we have marginal coverage $1 - \alpha$ under exchangeability.

### 4.1. Constructing estimators

We assume access to a set of calibration data $\{(X_i, Y_i)\}_{i=1}^n$ and a predictive model $f : \mathcal{X} \mapsto \Delta_{\mathcal{Y}}$. The output of the predictive model $f$ is a probability distribution over $\mathcal{Y}$ that estimates the true conditional distribution $P(y|x)$. We use the notation $f_x(y)$ to denote the probability assigned to label $y$ for the input $x$. Next, we develop an algorithm leveraging this model to take good actions based on a prediction set.

Following the oracle construction from Section 3, we first estimate $Q_t^x(a)$ using $f_x$: $\hat{Q}_t^x(a) := \inf\big\{\theta \in \mathbb{R} :$

---

**Algorithm 1** Risk–optimal conformal prediction (ROCP)

**Require:** Miscoverage level $\alpha \in (0,1)$, calibration samples $\{(X_i, Y_i)\}_{i=1}^n$, test covariate $X_{\text{test}}$.
1: **for** each candidate label $y \in \mathcal{Y}$ **do**
2:   Solve $\hat{\beta}_y \ := \ \arg\min_{\beta \geq 0} \beta$ such that

$$\frac{1}{n+1} \sum_{i=1}^n \mathbf{1}\{Y_i \in \hat{C}(X_i; \beta)\}$$
$$+ \frac{1}{n+1} \mathbf{1}\{y \in \hat{C}(X_{\text{test}}; \beta)\} \geq 1 - \alpha.$$

3: **end for**
**Ensure:** Prediction set $C_{\text{ROCP}}(X_{\text{test}}) \ := \ \{ y \in \mathcal{Y} : y \in \hat{C}(X_{\text{test}}; \hat{\beta}_y) \}$ and robust action $\hat{a}_{\text{ROCP}}(X_{\text{test}}) \in \arg\min_{a \in \mathcal{A}} L_{C_{\text{ROCP}}(X_{\text{test}})}(a; \alpha)$.

---

$\mathbb{P}_{Y \sim f_x}(\ell(a, Y) \leq \theta) \geq t\big\}$, then estimate $\hat{a}(x,t) \in \arg\min_{a \in \mathcal{A}} \big\{ t\hat{Q}_t^x(a) + (1-t)M(a) \big\}$, and set $\hat{\theta}(x,t) := \hat{Q}_t^x\big(\hat{a}(x,t)\big)$. Finally, define the empirical counterpart of $V_x$, $\hat{V}_x(t) := \min_{a \in \mathcal{A}} \big\{ t\,\hat{Q}_t^x(a) + (1-t)\,M(a) \big\}$, and the empirical dual selector: $\hat{g}(x,\beta) \in \arg\min_{t \in [0,1]} \big\{ \hat{V}_x(t) - \beta t \big\}$. From Theorem 3.3, we know that the optimal prediction sets admit a "one-dimensional" structure in terms of the scalar parameter $\beta \geq 0$, and the optimal coverage assignment is derived using the function $g$. Hence, to simplify notation, we define the $\beta$–*parametrized* quantities:

$$\hat{\theta}(x,\beta) \ := \ \hat{\theta}\big(x, \hat{g}(x,\beta)\big), \quad \hat{a}(x,\beta) \ := \ \hat{a}\big(x, \hat{g}(x,\beta)\big).$$

Following (11), the plug-in estimate of the optimal set takes the form $\hat{C}(x;\beta) := \hat{C}_0\big(x; \hat{g}(x,\beta)\big)$, where

$$\hat{C}_0(x;t) := \Big\{ y \in \mathcal{Y} : \ \ell\big(\hat{a}(x,t), y\big) \leq \hat{\theta}(x,t) \Big\}. \qquad (14)$$

### 4.2. Risk-optimal conformal prediction

We now give a simple finite-sample algorithm that we call risk-optimal conformal prediction (ROCP), in the spirit of RAC (Kiyani et al., 2025), that calibrates $\beta$ with distribution-free marginal coverage. See Algorithm 1. The algorithm only uses a calibration set and the maps $(\hat{\theta}, \hat{a}, \hat{g})$; it makes no assumptions about how $f$ was trained. Sections 2–3 are stated for general measurable outcome spaces, whereas Algorithm 1 is written for finite or discrete $\mathcal{Y}$, as done in our experiments.

**Proposition 4.1** (Distribution-free marginal coverage)**.** *Assume the calibration pairs $\{(X_i, Y_i)\}_{i=1}^n$ and the test pair $(X_{\text{test}}, Y_{\text{test}})$ are exchangeable. Then the ROCP set satisfies $\Pr\big\{ Y_{\text{test}} \in C_{\text{ROCP}}(X_{\text{test}}) \big\} \geq 1 - \alpha$, where the probability is over the joint randomness of the calibration and test data.*

*Remark* 4.2 (Coverage guarantee and risk targeting)**.** Proposition 4.1 gives a finite-sample, distribution-free marginal

---

[8]For simplicity in the main text, we assume that for every $\beta \geq 0$ the argmin in (12) is a singleton for $P_X$-a.e. $x$. The proof of Theorem 3.3 in Appendix C gives the general argument that allows pointwise ties.

[9]See Appendix C for the definition of normal integrands and its equivalent characterization.

coverage guarantee under exchangeability; this guarantee does not require the plug-in model $f_x$ to be correct. By contrast, the risk-targeting part of ROCP is model-dependent: the construction uses $f_x$ to estimate the oracle quantities $Q_t^x(a)$, $V_x(t)$, and $g(x, \beta)$. Thus, poor estimates can lead to suboptimal coverage allocation and weaker decision quality, even though the marginal coverage guarantee remains valid. If $\widehat{Q}$ is consistent and the argmins in $a$ and $t$ are stable, then we expect $(\widehat{\theta}, \widehat{a}, \widehat{g})$ to approximate $(\theta, a, g)$ well, and ROCP to approach the oracle risk performance in Theorem 3.3.

# 5. Experiments

Given any set $S \subseteq \mathcal{Y}$, define the robust decision rule $a_{\text{ROCP}}(S) \in \arg\min_{a \in \mathcal{A}} L_S(a; \alpha)$. In this section, we report experiments comparing our ROCP method with several baselines:

**Risk-Averse Calibration.** Kiyani et al. (2025) study a setting with a utility $u$, which is equivalent to the negative loss, $u = -\ell$. They propose the max–min decision rule $a_{\text{RA}}(C(x)) \in \arg\max_{a \in \mathcal{A}} \min_{y \in C(x)} u(a, y)$, which maps a prediction set $C(x)$ to an action by maximizing the worst-case utility over labels in the set. In our language, this corresponds to the rule $\arg\min_a \ell_{C(x)}^{\text{in}}(a)$, which corresponds to the solution $\pi^\star$ from (4) with $\alpha = 0$. They show that this rule is minimax-optimal for maximizing an expected quantile of the utility, and they derive prediction sets tailored to the max–min rule via their Algorithm 1. In our experiments, we evaluate both their full procedure RAC (Algorithm 1 paired with the max–min decision rule) and the max–min decision rule applied to prediction sets produced by other methods.

**Calibration + Best-Response.** We first calibrate the predictive model using *decision calibration* on the calibration set (Zhao et al., 2021). We then take the *best-response* action under the (approximately calibrated) predictive distribution, best-response$(x) \in \arg\min_{a \in \mathcal{A}} \mathbb{E}_{Y \sim f_x}[\ell(a, Y)]$. This baseline treats the model's predictive probabilities as reliable and commits to a single action without accounting for set-valued uncertainty. We include it to illustrate the consequences of fully trusting the model: while it can achieve strong average utility when the predictive distribution is accurate, it may also incur frequent critical mistakes compared to our method.

**Conformal Prediction.** As decision-agnostic baselines, we construct prediction sets with marginal $(1-\alpha)$-coverage via split conformal prediction using three scoring rules: Least Ambiguous Sets (LAS) (Sadinle et al., 2019), Adaptive Prediction Sets (APS) (Romano et al., 2020), and SOCOP (Wang et al., 2025). Given some $x$ and for each conformal set $C(x)$, we then instantiate downstream decisions using both our action rule $a_{\text{ROCP}}(C(x))$ and the risk-averse

max–min rule $a_{\text{RA}}(C(x))$ for comparison.

We evaluate the following metrics on the test set $\{(x_i, y_i)\}_{i=1}^{n_{\text{test}}}$, where each method outputs a prediction set $C(x_i)$ and a corresponding action $a_i$. (a) **Average Realized Worst-Case Risk**: the test-time mean of the worst-case robust risk $\frac{1}{n_{\text{test}}} \sum_{i=1}^{n_{\text{test}}} L_{C(x_i)}(a_i; \alpha)$, where $L_S(a; \alpha)$ is defined in Lemma 2.1 and $a_i$ is the chosen action. (b) **Average Realized Loss**: the test-time mean realized loss $\frac{1}{n_{\text{test}}} \sum_{i=1}^{n_{\text{test}}} \ell(a_i, y_i)$, where $y_i$ is the true label of $x_i$ and $a_i$ is the chosen action. (c) **Average Miscoverage**: the empirical miscoverage rate $\frac{1}{n_{\text{test}}} \sum_{i=1}^{n_{\text{test}}} \mathbf{1}\{y_i \notin C(x_i)\}$. (d) **Critical Mistake Rates**: For each critical label $y_c$, we report the fraction of test data with true label $y_c$ for which the chosen action attains the worst possible loss for that label:
$$\sum_{i:\, y_i = y_c} \mathbf{1}\Big\{a_i \in \arg\max_{a \in \mathcal{A}} \ell(a, y_c)\Big\}\Big/|\{i:\, y_i = y_c\}|.$$

## 5.1. Medical Diagnosis

First, in order to ensure a sufficiently detailed comparison with the RAC method of Kiyani et al. (2025), we start by comparing our method on a medical diagnosis example replicated from their paper. The data come from the COVID-19 Radiography Database (Chowdhury et al., 2020; Rahman et al., 2021), which contains chest X-ray images labeled into four categories: Normal, Pneumonia, COVID-19, and Lung Opacity. We randomly partition the dataset into training (70%), calibration (10%), and test (20%) splits. For the predictive model, we use an Inception-v3 architecture (Szegedy et al., 2015; 2016) initialized with ImageNet-pretrained weights and fine-tune it on the training split.

**Loss matrix designs.** We model the downstream treatment objective using a loss matrix[10] $\Lambda \in \mathbb{R}_+^{|\mathcal{Y}| \times |\mathcal{A}|}$, where $\ell(a, y) = \Lambda_{y,a}$. Here the labels are $y \in \{$ Normal, Pneumonia, COVID-19, Lung Opacity$\}$ and the available actions are $a \in \{$ No Action, Antibiotics, Quarantine, Additional Testing $\}$. Following Kiyani et al. (2025), we use the baseline matrix $\Lambda_0$, transformed by $\ell(a, y) = \max_{a \in \mathcal{A}, y \in \mathcal{Y}} u(a, y) - u(a, y)$ with their utility function $u$. To probe higher-stakes regimes in which incorrect interventions are substantially more costly, we additionally consider a variant $\Lambda_1$ where critical mistakes have higher losses:

$$\Lambda_0 = \begin{pmatrix} 0 & 8 & 8 & 6 \\ 10 & 0 & 7 & 3 \\ 10 & 7 & 0 & 2 \\ 9 & 6 & 6 & 0 \end{pmatrix}, \quad \Lambda_1 = \begin{pmatrix} 0 & 8 & 8 & 6 \\ 100 & 0 & 70 & 3 \\ 100 & 70 & 0 & 2 \\ 90 & 60 & 60 & 0 \end{pmatrix}.$$

This variant multiplies the loss of severe mismatches (e.g., choosing *No Action* for a pathological label) by a factor of 10, making it especially important to account for potentially out-of-set labels. This is the regime where ROCP's out-of-

---

[10]RAC is formulated in terms of a utility matrix. We convert losses to utilities via $u(a, y) = M - \ell(a, y)$, where $M = \max_{a \in \mathcal{A},\, y \in \mathcal{Y}} \ell(a, y)$.

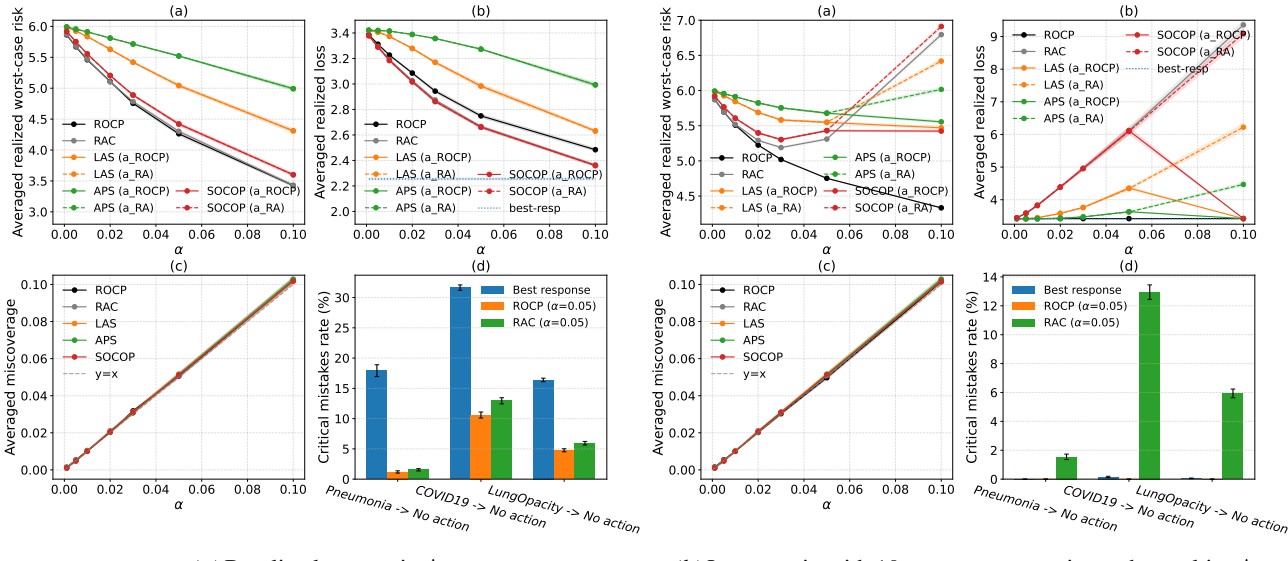

*(a)* Baseline loss matrix $\Lambda_0$           *(b)* Loss matrix with $10\times$ more severe-mismatch penalties $\Lambda_1$

*Figure 1.* Medical diagnosis experiments. Results under two treatment-loss specifications: the baseline loss matrix from Kiyani et al. (2025) (left) and our safety-critical variant. Each panel reports, as a function of miscoverage level $\alpha$: (a) average realized worst-case risk certificate; (b) average realized loss; (c) empirical miscoverage; (d) critical mistake rate for critical labels, defined as the fraction of test points with true label $y_c$ for which the chosen action attains $\arg\max_{a \in \mathcal{A}} \ell(a, y_c)$. All results are averaged over 20 random train/calibration/test splits; error bars show $\pm 1$ standard error.

set robustness can differ most from in-set max-min policies.

We vary the miscoverage level $\alpha$ to study its effect on performance. In Figure 1a, which uses the baseline loss matrix, ROCP attains worst-case risks that are close to RAC across all $\alpha$. While ROCP has slightly higher average realized loss than RAC, it consistently yields lower critical mistake rates. The best-response rule achieves the lowest average realized loss, but it has much higher critical mistake rates.

Finally, for decision-agnostic conformal set constructions (LAS/APS/SOCOP), applying $a_{\text{ROCP}}$ or $a_{\text{RA}}$ yields nearly identical performance, indicating that the dominant effect there comes from the set itself rather than the downstream policy. This close agreement is expected: under the baseline matrix there is no extreme penalty for severe mismatches, so the additional out-of-set term in the robust objective $L_{C(x)}(a; \alpha)$ (which is down-weighted by $\alpha$) has limited influence, and the resulting decision rule is close to the in-set max–min behavior of RAC.

The picture changes under the $10\times$ severe-mismatch penalty matrix in Figure 1b. In this higher-stakes regime, RAC becomes brittle as $\alpha$ grows: both its worst-case certificate and realized loss deteriorate sharply at larger $\alpha$, whereas ROCP remains stable and achieves substantially lower worst-case risk and realized loss, nearly matching the best-response baseline on realized loss. Moreover, ROCP almost eliminates critical mistakes for all critical labels while RAC remains similar to the baseline-matrix setting, highlighting the benefit of explicitly accounting for the $\alpha$ fraction of

out-of-set mass in the objective. This effect is also reflected in the conformal baselines: for $\alpha \geq 0.05$, applying $a_{\text{ROCP}}$ to LAS/APS/SOCOP sets consistently yields lower worst-case risk certificates and realized losses than applying the in-set max–min rule $a_{\text{RA}}$ to the same sets.

### 5.2. Decision-making in an "autonomous driving"-like setting

We consider a toy, static autonomous-driving-like decision experiment built from the BDD100K driving dataset (Yu et al., 2020). Our goal is not to evaluate closed-loop autonomous driving, but to stress-test decision-making under set-valued uncertainty using a *black-box* probabilistic model $f_x$ constructed from a pretrained YOLO11 detector (Jocher & Qiu, 2024). Each image $x$ is mapped to a hazard label $Y = (Y_a, Y_\ell, Y_r) \in \{0, 1\}^3$, where $Y_a$ indicates an occupied *ahead-close* region (person or vehicle), and $Y_\ell / Y_r$ indicate a *left-close/right-close* nearby vehicle. The precise region-of-interest definitions, label construction from BDD annotations, the construction of $f_x$, and the definitions of the actions and loss are deferred to Appendix D.1.

As shown in Figure 2, ROCP matches or outperforms all baselines in both worst-case risk certificate and realized loss across all $\alpha$ values. In particular, for the decision-agnostic conformal set constructions, pairing the same prediction sets with our robust decision rule $a_{\text{ROCP}}$ consistently improves performance over the in-set max–min rule $a_{\text{RA}}$ once $\alpha \geq 0.03$, yielding lower worst-case certificates and lower

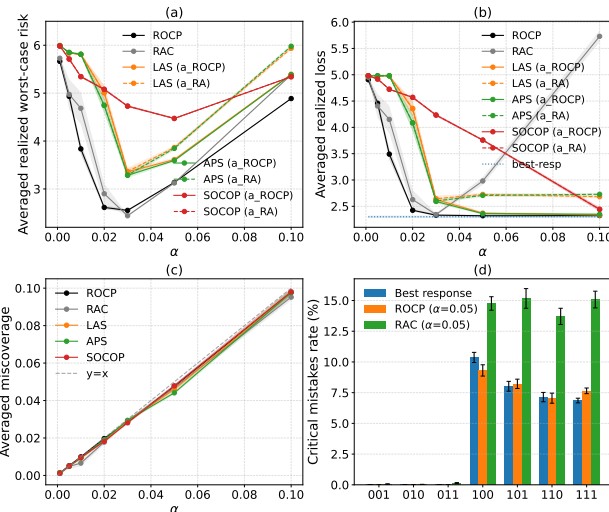

*Figure 2.* Toy autonomous driving experiment. (a) average realized worst-case risk certificate; (b) average realized loss; (c) empirical miscoverage; and (d) a critical mistake is defined as selecting an action that incurs the collision penalty (i.e., loss at least $M$) in that state. In (d), the x-axis labels 001, 010, ..., 111 denote the 3-bit hazard state $y = (y_a, y_\ell, y_r)$. Results are averaged over 20 random splits; error bars show $\pm 1$ standard error.

realized losses. Notably, the realized loss of RAC begins to increase after $\alpha \approx 0.03$, indicating that the max–min rule can become brittle when miscoverage is non-negligible in this setting, whereas ROCP remains stable.

Moreover, ROCP achieves lower critical mistake rates than RAC across hazardous states, while its realized loss remains close to that of RAC. This highlights the potential benefits of our method.

## 6. Discussion

Considering decision-making through the lens of expected loss minimization, we developed a decision-theoretic interface between conformal prediction sets and downstream action selection. Establishing formal approximation guarantees in our setting requires assumptions beyond exchangeability, including accurate estimation of the oracle quantities and stability of the relevant argmins, and we leave such analysis to future work. An important direction for future work is to incorporate group-conditional, label-conditional, or localized guarantees, which could further reduce brittle behavior on structured subpopulations when such guarantees are statistically feasible.

## Acknowledgments

This work was supported in part by the US NSF, ARO, AFOSR, ONR, the Simons Foundation and the Sloan Foundation. The authors thank Hamed Hassani, Shayan Kiyani, and Aaron Roth for helpful discussions about the work (Kiyani et al., 2025). They also thank Richard Berk and Nick Koning for helpful comments and references which have helped improve the paper.

## Impact Statement

This paper presents work whose goal is to advance the field of machine learning. There are many potential societal consequences of our work, none of which we feel must be specifically highlighted here.

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

## A. ROCP Workflow

# Risk-Optimal Conformal Prediction (ROCP)

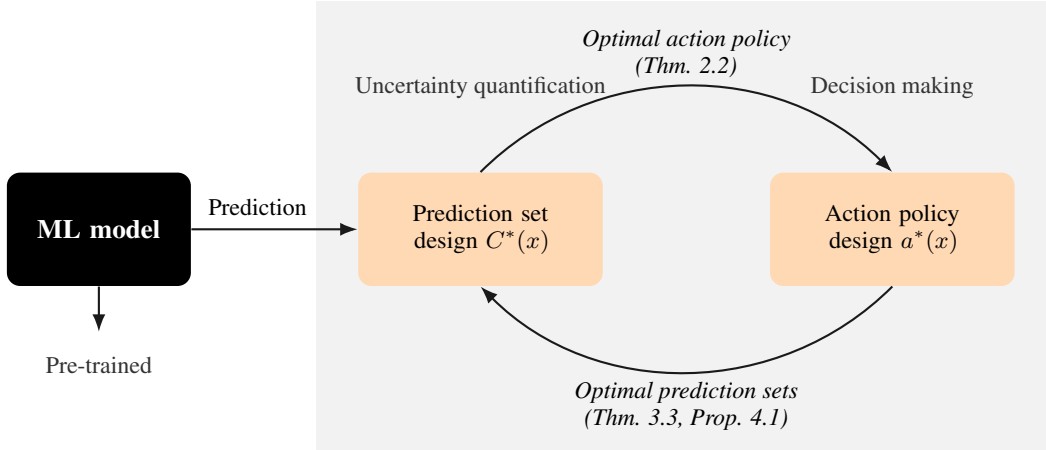

*Figure 3.* The pipeline of Risk-Optimal Conformal Prediction (ROCP).

## B. Additional Related Work and Background

**Conformal prediction and prediction sets.** Prediction sets have classical roots in statistics (Wilks, 1941; Wald, 1943; Scheffe & Tukey, 1945; Tukey, 1947) and were developed into the modern conformal prediction framework starting with Saunders et al. (1999); Vovk et al. (1999); Papadopoulos et al. (2002); Vovk et al. (2005). With the rise of modern ML, conformal prediction has become a standard tool for distribution-free uncertainty quantification across tasks including classification and regression (Lei et al., 2018; Romano et al., 2020; 2019; Park et al., 2022; Angelopoulos et al., 2020; 2023). Our work uses conformal prediction only for its coverage guarantee; the key question we address is how to turn that guarantee into *optimal downstream decisions* and *decision-optimal set construction*.

**Conformal methods for risk control and decision-aware sets.** A growing literature goes beyond coverage and uses conformal ideas to control general risk measures (Angelopoulos et al., 2022; 2021; Lindemann et al., 2023; Lekeufack et al., 2024; Zecchin & Simeone, 2024a; Blot et al., 2024; Zecchin & Simeone, 2024b; Cortes-Gomez et al., 2025; Joshi et al., 2025b). These works typically focus on guaranteeing that a chosen risk functional of the predictor is below a target level, often for a fixed decision rule. In contrast, our goal is to *jointly* characterize the optimal decision rule induced by coverage and the optimal prediction sets for that induced rule when the downstream objective is expected loss under worst-case distributions consistent with coverage.

**Certified decisions and econometric decision theory.** Our work is related to Andrews & Chen (2025), who study the connection between uncertainty quantification and downstream decision-making through the lens of certified decisions. Both papers connect uncertainty quantification to decisions, but they emphasize different objects and criteria. Andrews & Chen (2025) study P-certified decisions in a parameter-inference setting, i.e., recommended actions paired with high-probability upper bounds on loss, and show that under their dominance criterion, it is without loss to focus on certificates induced by minimax as-if decisions over confidence sets for an unknown parameter. This connects to the econometric perspective of as-if decisions with set estimates (Manski, 2021). By contrast, our paper studies prediction sets over labels, derives the minimax-optimal action for a fixed prediction set, characterizes the oracle set-design problem for a worst-case expected-loss objective, and proposes ROCP as a conformal algorithm targeting those sets. Put differently, their paper centers on certified decisions and the downstream adoption problem, whereas ours centers on the interface between prediction sets and actions together with decision-optimal set construction. The closest conceptual overlap is the minimax-over-set idea, whereas our robust expected-loss objective explicitly accounts for the unconstrained $\alpha$ mass outside the set. As a result, the optimal action in our framework is generally not the pure in-set max–min rule.

There are also connections to the broader econometrics literature on replacing infeasible conditional restrictions by feasible marginal or unconditional ones. For example, conditional moment restrictions can be converted into collections of

unconditional moment restrictions for inference (Andrews & Shi, 2013). Related conditional-to-marginal ideas have also appeared in conformal inference (Gibbs et al., 2025). Our oracle analysis uses conditional coverage to obtain a clean pointwise decision rule, while the practical ROCP algorithm provides finite-sample marginal coverage.

**Calibration and best-response baselines.** When probabilistic forecasts are calibrated, best-responding to the predictive distribution is optimal for expectation-maximizing agents (Zhao et al., 2021; Noarov et al., 2023). We include calibrated best-response baselines in our experiments to highlight the practical trade-off: committing to a single action can deliver strong average performance when the model is reliable, but can incur catastrophic errors when tail events are misestimated.

## C. Proofs

**Measurability conventions.** In the main text we suppress measure-theoretic details; throughout this appendix we make them explicit. Let $(\mathcal{X}, \mathcal{F})$, $(\mathcal{Y}, \mathcal{G})$, and $(\mathcal{A}, \mathcal{H})$ be standard Borel measurable spaces. All random variables are defined on a common probability space and take values in $\mathcal{X}$ and $\mathcal{Y}$. The loss $\ell : \mathcal{A} \times \mathcal{Y} \to [0, \infty)$ is assumed $\mathcal{H} \otimes \mathcal{G}$-measurable, and regular conditional laws exist. Whenever we write $Q(S)$ or $\Pr(Y \in S)$, we implicitly assume $S \in \mathcal{G}$ is measurable. For set-valued maps $C : \mathcal{X} \to 2^{\mathcal{Y}}$ appearing inside probabilities such as $\Pr\{Y \in C(X)\}$, we assume $C$ is measurable in the sense that its graph $\{(x, y) : y \in C(x)\}$ belongs to $\mathcal{F} \otimes \mathcal{G}$, so the event $\{Y \in C(X)\}$ is measurable. We also tacitly restrict to policies $\pi$ for which $x \mapsto \pi(C(x))$ is $\mathcal{H}$-measurable, ensuring $\ell(\pi(C(X)), Y)$ is measurable.

**Proof of Lemma 2.1.** Fix any measurable $S \subseteq \mathcal{Y}$ and $a \in \mathcal{A}$.

**Case 1:** $S = \mathcal{Y}$. The constraint $Q(\mathcal{Y}) \geq 1 - \alpha$ holds for any probability measure $Q$. Since $\mathbb{E}_{Y \sim Q}[\ell(a, Y)] \leq \sup_{y \in \mathcal{Y}} \ell(a, y)$, and conversely, taking $Q = \delta_{y^*}$ with $y^* = \arg\max_{y \in \mathcal{Y}} \ell(a, y)$ yields $L_{\mathcal{Y}}(a; \alpha) = \ell_{\mathcal{Y}}^{\text{in}}(a)$, matching (3) under the convention $\ell_{\mathcal{Y}}^{\text{out}}(a) := \ell_{\mathcal{Y}}^{\text{in}}(a)$.

**Case 2:** $S \subsetneq \mathcal{Y}$. For any probability measure $Q$ on $\mathcal{Y}$ with $Q(S) \geq 1 - \alpha$,

$$
\begin{aligned}
\mathbb{E}_{Y \sim Q}[\ell(a, Y)] &= \int_S \ell(a, y)\, Q(\mathrm{d}y) \;+\; \int_{S^c} \ell(a, y)\, Q(\mathrm{d}y) \\
&\leq Q(S) \sup_{y \in S} \ell(a, y) \;+\; (1 - Q(S)) \sup_{y \notin S} \ell(a, y) \\
&= Q(S)\, \ell_S^{\text{in}}(a) \;+\; (1 - Q(S))\, \ell_S^{\text{out}}(a).
\end{aligned}
$$

The right-hand side is affine in $Q(S) \in [1 - \alpha, 1]$, hence

$$
\mathbb{E}_{Y \sim Q}[\ell(a, Y)] \leq \max \left\{ \ell_S^{\text{in}}(a),\; (1 - \alpha)\ell_S^{\text{in}}(a) + \alpha\, \ell_S^{\text{out}}(a) \right\}.
$$

Taking the supremum over all such $Q$ gives

$$
L_S(a; \alpha) \;\leq\; \max \left\{ \ell_S^{\text{in}}(a),\; (1 - \alpha)\ell_S^{\text{in}}(a) + \alpha\, \ell_S^{\text{out}}(a) \right\}.
$$

For the reverse inequality, fix $\varepsilon > 0$. By definition of supremum, choose $y_i \in S$ such that $\ell(a, y_i) \geq \ell_S^{\text{in}}(a) - \varepsilon$. If $\ell_S^{\text{out}}(a) \leq \ell_S^{\text{in}}(a)$, take $Q_\varepsilon = \delta_{y_i}$; then $Q_\varepsilon(S) = 1$ and

$$
\mathbb{E}_{Y \sim Q_\varepsilon}[\ell(a, Y)] \geq \ell_S^{\text{in}}(a) - \varepsilon.
$$

If instead $\ell_S^{\text{out}}(a) > \ell_S^{\text{in}}(a)$, choose $y_o \notin S$ such that $\ell(a, y_o) \geq \ell_S^{\text{out}}(a) - \varepsilon$, and take $Q_\varepsilon = (1 - \alpha)\delta_{y_i} + \alpha\, \delta_{y_o}$; then $Q_\varepsilon(S) = 1 - \alpha$ and

$$
\mathbb{E}_{Y \sim Q_\varepsilon}[\ell(a, Y)] \geq (1 - \alpha)(\ell_S^{\text{in}}(a) - \varepsilon) + \alpha(\ell_S^{\text{out}}(a) - \varepsilon) = (1 - \alpha)\ell_S^{\text{in}}(a) + \alpha\, \ell_S^{\text{out}}(a) - \varepsilon.
$$

In either case,

$$
L_S(a; \alpha) \;\geq\; \max \left\{ \ell_S^{\text{in}}(a),\; (1 - \alpha)\ell_S^{\text{in}}(a) + \alpha\, \ell_S^{\text{out}}(a) \right\} - \varepsilon.
$$

Letting $\varepsilon \downarrow 0$ yields the matching lower bound. Combining with the upper bound,

$$
L_S(a; \alpha) = \max \left\{ \ell_S^{\text{in}}(a),\; (1 - \alpha)\ell_S^{\text{in}}(a) + \alpha\, \ell_S^{\text{out}}(a) \right\} = \ell_S^{\text{in}}(a) + \alpha\big(\ell_S^{\text{out}}(a) - \ell_S^{\text{in}}(a)\big)_+,
$$

as claimed. □

**Proof of Theorem 2.2.** Fix any policy $\pi$ and write $a(x) := \pi(C(x))$. We first show that for any fixed $\pi$,

$$\sup_{P \in \mathcal{P}_\alpha} \mathbb{E}_P\, \ell\big(\pi(C(X)), Y\big) \;=\; \sup_{x \in \mathcal{X}} L_{C(x)}\big(\pi(C(x)); \alpha\big). \tag{15}$$

*Upper bound.* Take any $P \in \mathcal{P}_\alpha$. By the definition of $\mathcal{P}_\alpha$, for $P_X$-a.e. $x$ the conditional law $Q_x(\cdot) := P(Y \in \cdot \mid X = x)$ satisfies $Q_x(C(x)) \geq 1 - \alpha$. Hence, by the definition (2) of $L_S(a; \alpha)$,

$$\mathbb{E}_P\big[\ell\big(a(X), Y\big) \mid X = x\big] \leq L_{C(x)}\big(a(x); \alpha\big) = L_{C(x)}\big(\pi(C(x)); \alpha\big) \quad \text{for } P_X\text{-a.e. } x.$$

Taking expectations over $X$ gives

$$\mathbb{E}_P\, \ell\big(\pi(C(X)), Y\big) \leq \mathbb{E}_{X \sim P_X}\big[L_{C(X)}\big(\pi(C(X)); \alpha\big)\big] \leq \sup_{x \in \mathcal{X}} L_{C(x)}\big(\pi(C(x)); \alpha\big).$$

Since this holds for all $P \in \mathcal{P}_\alpha$, we obtain

$$\sup_{P \in \mathcal{P}_\alpha} \mathbb{E}_P\, \ell\big(\pi(C(X)), Y\big) \leq \sup_{x \in \mathcal{X}} L_{C(x)}\big(\pi(C(x)); \alpha\big).$$

*Lower bound.* Fix $\varepsilon > 0$ and choose $x_\varepsilon \in \mathcal{X}$ such that

$$L_{C(x_\varepsilon)}\big(\pi(C(x_\varepsilon)); \alpha\big) \;\geq\; \sup_{x \in \mathcal{X}} L_{C(x)}\big(\pi(C(x)); \alpha\big) - \varepsilon.$$

Let $S_\varepsilon := C(x_\varepsilon)$ and $a_\varepsilon := \pi(S_\varepsilon)$, and write $\ell_\varepsilon^{\mathrm{in}} := \ell_{S_\varepsilon}^{\mathrm{in}}(a_\varepsilon)$, $\ell_\varepsilon^{\mathrm{out}} := \ell_{S_\varepsilon}^{\mathrm{out}}(a_\varepsilon)$. Set $P_X = \delta_{x_\varepsilon}$.

*Case 1:* $S_\varepsilon \neq \mathcal{Y}$ and $\ell_\varepsilon^{\mathrm{out}} > \ell_\varepsilon^{\mathrm{in}}$. Choose $y_{\mathrm{i}} \in S_\varepsilon$ and $y_{\mathrm{o}} \notin S_\varepsilon$ such that

$$\ell(a_\varepsilon, y_{\mathrm{i}}) \geq \ell_\varepsilon^{\mathrm{in}} - \varepsilon, \qquad \ell(a_\varepsilon, y_{\mathrm{o}}) \geq \ell_\varepsilon^{\mathrm{out}} - \varepsilon.$$

Define $Y \mid X = x_\varepsilon \sim (1 - \alpha)\delta_{y_{\mathrm{i}}} + \alpha\, \delta_{y_{\mathrm{o}}}$. Then $P \in \mathcal{P}_\alpha$ and

$$\mathbb{E}_P\, \ell\big(\pi(C(X)), Y\big) \geq (1 - \alpha)\ell_\varepsilon^{\mathrm{in}} + \alpha\, \ell_\varepsilon^{\mathrm{out}} - \varepsilon = L_{S_\varepsilon}(a_\varepsilon; \alpha) - \varepsilon.$$

*Case 2:* $S_\varepsilon = \mathcal{Y}$ or $\ell_\varepsilon^{\mathrm{out}} \leq \ell_\varepsilon^{\mathrm{in}}$. Choose $y_{\mathrm{i}} \in S_\varepsilon$ such that $\ell(a_\varepsilon, y_{\mathrm{i}}) \geq \ell_\varepsilon^{\mathrm{in}} - \varepsilon$ and define $Y \mid X = x_\varepsilon \sim \delta_{y_{\mathrm{i}}}$. Then $P \in \mathcal{P}_\alpha$ and

$$\mathbb{E}_P\, \ell\big(\pi(C(X)), Y\big) = \ell(a_\varepsilon, y_{\mathrm{i}}) \geq \ell_\varepsilon^{\mathrm{in}} - \varepsilon = L_{S_\varepsilon}(a_\varepsilon; \alpha) - \varepsilon.$$

Combining the two cases, we have constructed $P \in \mathcal{P}_\alpha$ such that

$$\mathbb{E}_P\, \ell\big(\pi(C(X)), Y\big) \;\geq\; L_{S_\varepsilon}(a_\varepsilon; \alpha) - \varepsilon.$$

By the choice of $x_\varepsilon$ and since $S_\varepsilon = C(x_\varepsilon)$, also

$$L_{S_\varepsilon}(a_\varepsilon; \alpha) = L_{C(x_\varepsilon)}\big(\pi(C(x_\varepsilon)); \alpha\big) \geq \sup_{x \in \mathcal{X}} L_{C(x)}\big(\pi(C(x)); \alpha\big) - \varepsilon.$$

Therefore

$$\sup_{P \in \mathcal{P}_\alpha} \mathbb{E}_P\, \ell\big(\pi(C(X)), Y\big) \;\geq\; \sup_{x \in \mathcal{X}} L_{C(x)}\big(\pi(C(x)); \alpha\big) - 2\varepsilon.$$

Letting $\varepsilon \downarrow 0$ yields the reverse inequality in (15).

Now minimize over $\pi$. For any $\pi$,

$$\sup_{x \in \mathcal{X}} L_{C(x)}\big(\pi(C(x)); \alpha\big) \;\geq\; \sup_{x \in \mathcal{X}} \min_{a \in \mathcal{A}} L_{C(x)}(a; \alpha),$$

hence

$$\inf_\pi \sup_{P \in \mathcal{P}_\alpha} \mathbb{E}\, \ell\big(\pi(C(X)), Y\big) = \inf_\pi \sup_{x \in \mathcal{X}} L_{C(x)}\big(\pi(C(x)); \alpha\big) \;\geq\; \sup_{x \in \mathcal{X}} \min_{a \in \mathcal{A}} L_{C(x)}(a; \alpha).$$

Conversely, by the attainment assumption, for each set $S$ in the range $\mathrm{Im}(C) = \{C(x) : x \in \mathcal{X}\}$ pick $\pi^\star(S) \in \arg\min_{a \in \mathcal{A}} L_S(a; \alpha)$. Then

$$\sup_{x \in \mathcal{X}} L_{C(x)}\big(\pi^\star(C(x)); \alpha\big) = \sup_{x \in \mathcal{X}} \min_{a \in \mathcal{A}} L_{C(x)}(a; \alpha),$$

so equality holds in (5) and $\pi^\star$ satisfies (4).

Finally, under the additional attainment assumptions of the relevant suprema in $\ell^{\mathrm{in}}$ and $\ell^{\mathrm{out}}$, and attainment of the outer supremum over $x$, the above construction with $\varepsilon = 0$ yields a worst-case $P^\star \in \mathcal{P}_\alpha$ concentrated at such an $x^\star$, with $Y \mid X = x^\star$ as stated in the theorem. $\qquad\square$

**Details for Remark 2.3.** Fix a prediction set $C$ and a policy $\pi$, and define

$$\bar{\ell}^{\mathrm{in}}_\pi := \sup_x \ell^{\mathrm{in}}_{C(x)}(\pi(C(x))), \qquad \bar{\ell}^{\mathrm{out}}_\pi := \sup_x \ell^{\mathrm{out}}_{C(x)}(\pi(C(x))).$$

For the marginal ambiguity class

$$\mathcal{P}^{\mathrm{m}}_\alpha(C) := \{P \text{ on } \mathcal{X} \times \mathcal{Y} : \Pr(Y \in C(X)) \geq 1 - \alpha\},$$

the same argument as in Lemma 2.1, with the in-set and out-of-set suprema now taken over all covariates, gives

$$\sup_{P \in \mathcal{P}^{\mathrm{m}}_\alpha(C)} \mathbb{E}_P[\ell(\pi(C(X)), Y)] = \bar{\ell}^{\mathrm{in}}_\pi + \alpha\big(\bar{\ell}^{\mathrm{out}}_\pi - \bar{\ell}^{\mathrm{in}}_\pi\big)_+.$$

Indeed, every $P \in \mathcal{P}^{\mathrm{m}}_\alpha(C)$ satisfies

$$\mathbb{E}_P[\ell(\pi(C(X)), Y)] \leq \Pr(Y \in C(X))\bar{\ell}^{\mathrm{in}}_\pi + \Pr(Y \notin C(X))\bar{\ell}^{\mathrm{out}}_\pi,$$

and maximizing the right-hand side subject to $\Pr(Y \notin C(X)) \leq \alpha$ gives the displayed upper bound. The matching lower bound follows by putting mass on covariate values and labels that approximate the corresponding worst in-set and out-of-set losses, as in the lower-bound argument for Lemma 2.1.

Now let

$$R_{\mathrm{cond}}(\pi) := \sup_x L_{C(x)}(\pi(C(x)); \alpha), \qquad R_{\mathrm{marg}}(\pi) := \sup_{P \in \mathcal{P}^{\mathrm{m}}_\alpha(C)} \mathbb{E}_P[\ell(\pi(C(X)), Y)].$$

Then

$$R_{\mathrm{cond}}(\pi) \leq R_{\mathrm{marg}}(\pi) \leq (2 - \alpha)R_{\mathrm{cond}}(\pi).$$

The first inequality follows by comparing the pointwise and global suprema. For the second, note that $R_{\mathrm{cond}}(\pi) \geq \bar{\ell}^{\mathrm{in}}_\pi$ and $R_{\mathrm{cond}}(\pi) \geq \alpha\bar{\ell}^{\mathrm{out}}_\pi$. If $\bar{\ell}^{\mathrm{out}}_\pi > \bar{\ell}^{\mathrm{in}}_\pi$, then

$$R_{\mathrm{marg}}(\pi) = (1 - \alpha)\bar{\ell}^{\mathrm{in}}_\pi + \alpha\bar{\ell}^{\mathrm{out}}_\pi \leq (1 - \alpha)R_{\mathrm{cond}}(\pi) + R_{\mathrm{cond}}(\pi) = (2 - \alpha)R_{\mathrm{cond}}(\pi).$$

If $\bar{\ell}^{\mathrm{out}}_\pi \leq \bar{\ell}^{\mathrm{in}}_\pi$, then $R_{\mathrm{marg}}(\pi) = \bar{\ell}^{\mathrm{in}}_\pi \leq R_{\mathrm{cond}}(\pi)$.

Let $\pi_{\mathrm{cond}}$ minimize $R_{\mathrm{cond}}$, and let $\pi_{\mathrm{marg}}$ minimize $R_{\mathrm{marg}}$. Then

$$R_{\mathrm{marg}}(\pi_{\mathrm{cond}}) \leq (2 - \alpha)R_{\mathrm{cond}}(\pi_{\mathrm{cond}}) \leq (2 - \alpha)R_{\mathrm{cond}}(\pi_{\mathrm{marg}}) \leq (2 - \alpha)R_{\mathrm{marg}}(\pi_{\mathrm{marg}}).$$

Thus, for the same fixed prediction set $C$, the policy from Theorem 2.2 is a $(2 - \alpha)$-approximation to the optimal marginal policy. $\qquad\square$

*Remark* C.1 (Measurable selection for (4)). The final step of the proof above picks, for each set $S$ in the range of $C$, some minimizer $\pi^\star(S) \in \arg\min_{a \in \mathcal{A}} L_S(a; \alpha)$. This pointwise choice does not automatically ensure that the composite map $x \mapsto \pi^\star(C(x))$ is $\mathcal{H}$-measurable, as required by our measurability conventions. A sufficient condition is the following: assume that the function $f : \mathcal{X} \times \mathcal{A} \to \overline{\mathbb{R}}$ defined by $f(x, a) := L_{C(x)}(a; \alpha)$ is a normal integrand in the sense of Definition C.2 and that, for every $x$, the minimum of $a \mapsto f(x, a)$ is attained. Then the argmin correspondence $\Gamma(x) := \arg\min_{a \in \mathcal{A}} f(x, a)$ is a measurable multifunction with nonempty closed values (see, e.g., (Rockafellar & Wets, 1998), Thm. 14.37). Since $(\mathcal{A}, \mathcal{H})$ is standard Borel, the Kuratowski–Ryll–Nardzewski measurable selection theorem (Kuratowski & Ryll-Nardzewski, 1965) yields an $\mathcal{H}$-measurable selector $a^\star : \mathcal{X} \to \mathcal{A}$ such that $a^\star(x) \in \Gamma(x)$ for all $x$. Taking $x \mapsto \pi^\star(C(x)) := a^\star(x)$ gives a measurable minimax-optimal policy satisfying (4).

**Proof of Proposition 3.1.** Fix $x \in \mathcal{X}$ and $t \in (0, 1]$. Let $C \subseteq \mathcal{Y}$ be measurable with $\mathbb{P}(Y \in C \mid X = x) \geq t$, and fix any $a \in \mathcal{A}$. Set $s := \ell_C^{\mathrm{in}}(a) = \sup_{y \in C} \ell(a, y)$. Then $C \subseteq S_s(a) := \{y : \ell(a, y) \leq s\}$, hence

$$\mathbb{P}\big(\ell(a, Y) \leq s \mid X = x\big) = \mathbb{P}\big(Y \in S_s(a) \mid X = x\big) \geq \mathbb{P}\big(Y \in C \mid X = x\big) \geq t.$$

By definition of $Q_t^x(a)$, this implies $s \geq Q_t^x(a)$.

If $s < M(a)$, then $C^c \supseteq \{y : \ell(a, y) > s\}$, so $\ell_C^{\mathrm{out}}(a) = M(a)$. Applying Lemma 2.1 with $\alpha = 1 - t$ gives

$$L_C(a; 1 - t) = s + (1 - t)\big(\ell_C^{\mathrm{out}}(a) - s\big)_+ = s + (1 - t)(M(a) - s) = t\,s + (1 - t)M(a) \geq t\,Q_t^x(a) + (1 - t)M(a).$$

If $s = M(a)$, then $L_C(a; 1 - t) \geq s = M(a) \geq t\,Q_t^x(a) + (1 - t)M(a)$ (since $Q_t^x(a) \leq M(a)$). Therefore, for every feasible $C$,

$$R(C, 1 - t) = \min_{a \in \mathcal{A}} L_C(a; 1 - t) \geq \min_{a \in \mathcal{A}} \Big\{ t\,Q_t^x(a) + (1 - t)M(a) \Big\}.$$

Now let $a(x, t)$ be as in (7), define $\theta(x, t) := Q_t^x(a(x, t))$, and set $C(x, t)$ as in (8). Then $\mathbb{P}(Y \in C(x, t) \mid X = x) = \mathbb{P}(\ell(a(x, t), Y) \leq \theta(x, t) \mid X = x) \geq t$, so $C(x, t)$ is feasible. Moreover, since $C(x, t) = S_{\theta(x, t)}(a(x, t))$, we have $\ell_{C(x,t)}^{\mathrm{in}}(a(x, t)) \leq \theta(x, t)$. On the other hand, the argument above showed that for any feasible $C$, $\ell_C^{\mathrm{in}}(a) \geq Q_t^x(a)$; applying this to $C(x, t)$ and $a(x, t)$ yields

$$\ell_{C(x,t)}^{\mathrm{in}}(a(x, t)) \geq Q_t^x(a(x, t)) = \theta(x, t).$$

Therefore $\ell_{C(x,t)}^{\mathrm{in}}(a(x, t)) = \theta(x, t)$. If $\theta(x, t) < M(a(x, t))$, then $\ell_{C(x,t)}^{\mathrm{out}}(a(x, t)) = M(a(x, t))$. If instead $\theta(x, t) = M(a(x, t))$, then trivially $\ell_{C(x,t)}^{\mathrm{out}}(a(x, t)) \leq M(a(x, t)) = \ell_{C(x,t)}^{\mathrm{in}}(a(x, t))$, so $\big(\ell_{C(x,t)}^{\mathrm{out}}(a(x, t)) - \ell_{C(x,t)}^{\mathrm{in}}(a(x, t))\big)_+ = 0$. Thus Lemma 2.1 yields

$$L_{C(x,t)}(a(x, t); 1 - t) = t\,\theta(x, t) + (1 - t)M(a(x, t)).$$

Consequently,

$$R(C(x, t), 1 - t) \leq L_{C(x,t)}(a(x, t); 1 - t) = \min_{a \in \mathcal{A}} \Big\{ t\,Q_t^x(a) + (1 - t)M(a) \Big\},$$

which matches the lower bound, proving optimality and the claimed value.

Finally, if the suprema are attained, pick $y_{\mathrm{i}} \in \arg\max_{y \in C(x,t)} \ell(a(x, t), y)$ and $y_{\mathrm{o}} \in \arg\max_{y \in \mathcal{Y}} \ell(a(x, t), y)$ and take $Q^\star = t\delta_{y_{\mathrm{i}}} + (1 - t)\delta_{y_{\mathrm{o}}}$. Then $Q^\star(C(x, t)) \geq t$ and $\mathbb{E}_{Y \sim Q^\star}[\ell(a(x, t), Y)] = t\,\theta(x, t) + (1 - t)M(a(x, t)) = L_{C(x,t)}(a(x, t); 1 - t)$, so $Q^\star$ achieves the inner supremum in $L_{C(x,t)}(a(x, t); 1 - t)$. $\qquad\square$

**Proof of Theorem 3.3.** We first introduce the definition of normal integrands.

**Definition C.2** (Normal integrands (Rockafellar & Wets, 1998))**.** Let $(\mathcal{X}, \mathcal{F})$ be a measurable space and let $(\mathcal{Z}, \mathcal{B}(\mathcal{Z}))$ be a Polish space with its Borel $\sigma$-field. A function $f : \mathcal{X} \times \mathcal{Z} \to \overline{\mathbb{R}}$ is called a *normal integrand* if its epigraphical mapping $S_f : \mathcal{X} \rightrightarrows \mathcal{Z} \times \mathbb{R}$, defined by

$$S_f(x) := \mathrm{epi}\, f(x, \cdot) := \big\{ (z, \alpha) \in \mathcal{Z} \times \mathbb{R} : f(x, z) \leq \alpha \big\},$$

is closed-valued and measurable (i.e., its graph $\{(x, z, \alpha) : (z, \alpha) \in S_f(x)\}$ belongs to $\mathcal{F} \otimes \mathcal{B}(\mathcal{Z}) \otimes \mathcal{B}(\mathbb{R})$).

Take $\mathcal{Z} = [0, 1]$. We assume $(x, t) \mapsto V_x(t)$ in (10) is a normal integrand in the sense of Definition C.2. Equivalently,[11]

(i) $(x, t) \mapsto V_x(t)$ is $\mathcal{F} \otimes \mathcal{B}([0, 1])$-measurable;

(ii) for each $x$, the map $t \mapsto V_x(t)$ is lower semicontinuous on $[0, 1]$.

Recall the population problem (9):

$$\mathrm{VAL}(\alpha) = \inf_{\substack{t : \mathcal{X} \to [0, 1]\text{ measurable} \\ \mathbb{E}[t(X)] \geq 1 - \alpha}} \mathbb{E}\big[ V_X\big(t(X)\big) \big].$$

---

[11] Under the measurability conventions, a sufficient condition is that $\mathcal{A}$ is a compact metric space with $\mathcal{H} = \mathcal{B}(\mathcal{A})$ and that $a \mapsto \ell(a, y)$ is lower semicontinuous for every $y \in \mathcal{Y}$.

For $\beta \geq 0$, define the dual function

$$\phi(\beta) := \beta(1 - \alpha) + \mathbb{E}\left[\inf_{u \in [0,1]} \{V_X(u) - \beta u\}\right].$$

and define the (set-valued) argmin correspondence

$$\Gamma_\beta(x) := \arg\min_{u \in [0,1]} \{V_x(u) - \beta u\},$$

and its extremal selectors

$$g^+(x, \beta) := \max \Gamma_\beta(x), \qquad g^-(x, \beta) := \min \Gamma_\beta(x).$$

Since we assume that $(x, u) \mapsto V_x(u)$ is a normal integrand and $u \mapsto -\beta u$ is continuous, the function $V_x(u) - \beta u$ is also a normal integrand; hence $\Gamma_\beta(x) = \arg\min_{u \in [0,1]} \{V_x(u) - \beta u\}$ is a measurable multifunction with nonempty compact values (see, e.g., (Rockafellar & Wets, 1998), Thm. 14.37), and therefore $g^-(\cdot, \beta)$ and $g^+(\cdot, \beta)$ are measurable (see, e.g., (Rockafellar & Wets, 1998), Def. 14.1 or Ex. 14.51).

We prove a general statement of Theorem 3.3 where we do not assume pointwise uniqueness of the minimizer in (12).

**Theorem C.3.** *Assume $P_X$ is non-atomic, that $(x, t) \mapsto V_x(t)$ in (10) is a normal integrand, and that, for each $x$, the minimum in the definition of $V_x(t)$ exists for every $t \in [0, 1]$. Then there exists $\beta^* \geq 0$ and a measurable $t^\star : \mathcal{X} \to [0, 1]$ such that*

$$t^\star(x) \in \Gamma_{\beta^\star}(x) \quad P_X\text{-a.e. } x,$$

*and $t^*$ solves the population problem (9). Consequently, the optimal prediction sets are obtained using $t^\star(x)$ from (11), with actions $a^\star(x) = a(x, t^\star(x))$ as in (7). Moreover, one can always choose $t^*$ of the form*

$$t^*(x) = \begin{cases} g^-(x, \beta^\star) + (g^+(x, \beta^\star) - g^-(x, \beta^\star))\mathbf{1}_A(x), & \beta^* > 0, \\ g^+(x, 0), & \beta^* = 0. \end{cases} \tag{16}$$

*for some measurable $A \subseteq \mathcal{X}$. Any maximizer $\beta^\star > 0$ of the dual function $\phi(\beta)$ satisfies the interval condition*

$$\mathbb{E}\left[g^-(X, \beta^\star)\right] \leq 1 - \alpha \leq \mathbb{E}\left[g^+(X, \beta^\star)\right]. \tag{17}$$

*If $\beta^\star = 0$, only the right inequality is required, i.e. $\mathbb{E}[g^+(X, 0)] \geq 1 - \alpha$.*

*Proof of Theorem C.3.* For $\beta \geq 0$, define the Lagrangian

$$\mathcal{L}(t; \beta) := \mathbb{E}\left[V_X(t(X))\right] + \beta\Big((1 - \alpha) - \mathbb{E}[t(X)]\Big) = \beta(1 - \alpha) + \mathbb{E}\left[V_X(t(X)) - \beta\, t(X)\right].$$

By Lemma C.4,

$$\inf_{\substack{t: \mathcal{X} \to [0,1] \\ t \text{ measurable}}} \mathcal{L}(t; \beta) = \beta(1 - \alpha) + \mathbb{E}\left[\inf_{u \in [0,1]} \{V_X(u) - \beta u\}\right] =: \phi(\beta).$$

Weak duality gives $\text{VAL}(\alpha) \geq \sup_{\beta \geq 0} \phi(\beta)$. By Lemma C.5, there is no duality gap, hence

$$\text{VAL}(\alpha) = \sup_{\beta \geq 0} \phi(\beta).$$

Moreover, the supremum is attained. For any $\beta \geq 0$,

$$\phi(\beta) = \beta(1 - \alpha) + \mathbb{E}\left[\inf_{u \in [0,1]} \{V_X(u) - \beta u\}\right] \leq \beta(1 - \alpha) + \mathbb{E}[V_X(1) - \beta] = \mathbb{E}[V_X(1)] - \alpha\beta,$$

so $\phi(\beta) \to -\infty$ as $\beta \to \infty$ (since $\alpha > 0$). Hence there exists $B < \infty$ such that $\sup_{\beta \geq 0} \phi(\beta) = \sup_{\beta \in [0, B]} \phi(\beta)$. Next, define $\psi_x(\beta) := \inf_{u \in [0,1]} \{V_x(u) - \beta u\}$. For any $\beta, \beta' \geq 0$ and any $x$,

$$|\psi_x(\beta) - \psi_x(\beta')| \leq |\beta - \beta'|$$

since $u \in [0, 1]$. Taking expectations and adding the linear term $\beta(1 - \alpha)$ shows that $\phi$ is Lipschitz (hence continuous) on $[0, B]$. Therefore, by compactness of $[0, B]$, there exists $\beta^\star \in [0, B]$ maximizing $\phi$. In addition, since each $\psi_x$ is concave and 1-Lipschitz in $\beta$, hence the one-sided derivatives $\psi'_{x,+}(\beta)$ and $\psi'_{x,-}(\beta)$ exist for all $\beta \geq 0$. By Danskin's theorem (Danskin, 2012), the one-sided derivatives satisfy

$$\psi'_{x,+}(\beta) = -g^+(x, \beta), \qquad \psi'_{x,-}(\beta) = -g^-(x, \beta).$$

Moreover, for any $h \neq 0$ we have

$$\left| \frac{\psi_x(\beta + h) - \psi_x(\beta)}{h} \right| \leq 1,$$

so by dominated convergence we can obtain the one-sided derivatives of $\phi$:

$$\phi'_+(\beta) = (1 - \alpha) + \mathbb{E}\big[\psi'_{X,+}(\beta)\big] = (1 - \alpha) - \mathbb{E}\big[g^+(X, \beta)\big],$$

$$\phi'_-(\beta) = (1 - \alpha) + \mathbb{E}\big[\psi'_{X,-}(\beta)\big] = (1 - \alpha) - \mathbb{E}\big[g^-(X, \beta)\big].$$

Since $\phi$ is concave on $[0, \infty)$ and $\beta^\star$ maximizes $\phi$ if and only if $0 \in \partial\phi(\beta^\star)$, where the superdifferential $\partial\phi(\beta)$ is the closed interval $[\phi'_+(\beta), \phi'_-(\beta)]$. Thus if $\beta^\star > 0$ (interior maximizer) then $\phi'_+(\beta^\star) \leq 0 \leq \phi'_-(\beta^\star)$, which is equivalent to the interval condition (17); For the boundary case $\beta^\star = 0$, view $\phi$ as an extended-real concave function on $\mathbb{R}$ by setting $\phi(\beta) := -\infty$ for $\beta < 0$. Then $\partial\phi(0) = [\phi'_+(0), +\infty)$, so the optimality condition $0 \in \partial\phi(0)$ reduces to $\phi'_+(0) \leq 0$, i.e. $1 - \alpha \leq \mathbb{E}[g^+(X, 0)]$.

Next, we construct a primal optimizer. If $\beta^\star = 0$, let $t^\star(x) := g^+(x, 0) \in \Gamma_0(x)$. Since in this case $1 - \alpha \leq \mathbb{E}[g^+(X, 0)]$, so $t^\star$ is measurable and feasible. If $\beta^\star > 0$, define $w(x) := g^+(x, \beta^\star) - g^-(x, \beta^\star) \geq 0$ and $r := (1 - \alpha) - \mathbb{E}[g^-(X, \beta^\star)]$. By the interval condition, $r \in [0, \mathbb{E}[w(X)]]$. Define the finite measure $\nu$ on $(\mathcal{X}, \mathcal{F})$ by

$$\nu(B) := \mathbb{E}\big[w(X)\mathbf{1}_{\{X \in B\}}\big] = \int_B w(x)P_X(dx).$$

Because $P_X$ is non-atomic and $\nu \ll P_X$, $\nu$ is also non-atomic. By Sierpiński's theorem (Sierpiński, 1922), for any $0 \leq s \leq \nu(\mathcal{X})$, there exists a measurable $A \in \mathcal{F}$ with $\nu(A) = s$. Apply this with $s = r$ to get $A \in \mathcal{F}$ such that $\mathbb{E}[w(X)\mathbf{1}_A(X)] = r$. Let

$$t^\star(x) := g^-(x, \beta^\star) + w(x)\mathbf{1}_A(x).$$

Then $t^\star(x) \in \{g^-(x, \beta^\star), g^+(x, \beta^\star)\} \subseteq \Gamma_{\beta^\star}(x)$, and

$$\mathbb{E}[t^\star(X)] = \mathbb{E}[g^-(X, \beta^\star)] + \mathbb{E}[w(X)\mathbf{1}_A(X)] = 1 - \alpha.$$

Finally, since in both case, $t^\star(x) \in \Gamma_{\beta^\star}(x)$,

$$V_X(t^\star(X)) - \beta^\star t^\star(X) = \inf_{u \in [0,1]} \{V_X(u) - \beta^\star u\} \quad \text{a.s.}$$

Taking expectations gives

$$\mathbb{E}[V_X(t^\star(X))] + \beta^\star\big((1 - \alpha) - \mathbb{E}[t^\star(X)]\big) = \phi(\beta^\star),$$

If $\beta^\star > 0$, we constructed $\mathbb{E}[t^\star(X)] = 1 - \alpha$, so the Lagrange term vanishes and $\mathbb{E}[V_X(t^\star(X))] = \phi(\beta^\star)$. If $\beta^\star = 0$, then $\mathbb{E}[V_X(t^\star(X))] = \phi(0)$ directly. In both cases, since $\beta^\star$ maximizes $\phi$,

$$\mathbb{E}[V_X(t^\star(X))] = \phi(\beta^\star) = \max_{\beta \geq 0} \phi(\beta) = \text{VAL}(\alpha)$$

by strong duality (Lemma C.5). Hence, $t^\star$ is primal optimal.

Finally, plugging $t^\star$ into (11) yields the stated optimal prediction sets, with actions $a^\star(x) = a\big(x, t^\star(x)\big)$ as in (7). $\qquad \square$

*Proof of Theorem 3.3.* In the main text we assume that for every $\beta \geq 0$, the argmin in (12) is a singleton for $P_X$-a.e. $x$. Thus, for each fixed $\beta$, $\Gamma_\beta(x) = \{g(x, \beta)\}$ for $P_X$-a.e. $x$, and

$$g^-(x, \beta) = g^+(x, \beta) = g(x, \beta), \text{ a.s.}$$

Then our general form (16) reduces to $t^\star(x) = g(x, \beta^\star)$.

Moreover, by Theorem C.3, any maximizer $\beta^\star > 0$ satisfies the interval condition $\mathbb{E}[g^-(X, \beta^\star)] \leq 1-\alpha \leq \mathbb{E}[g^+(X, \beta^\star)]$. Under the singleton assumption, $g^-(X, \beta^\star) = g^+(X, \beta^\star) = g(X, \beta^\star)$ almost surely, hence the interval collapses to

$$\mathbb{E}\big[g(X, \beta^\star)\big] = 1 - \alpha,$$

which is exactly (13).

Conversely, suppose some $\beta \geq 0$ solves (13), and define $t_\beta(x) := g(x, \beta)$. Then $t_\beta(x) \in \Gamma_\beta(x)$ for $P_X$-a.e. $x$ and $\mathbb{E}[t_\beta(X)] = 1 - \alpha$. From the proof of Theorem C.3, the dual function $\phi$ is concave and its one-sided derivatives satisfy

$$\phi'_+(\beta) = (1-\alpha) - \mathbb{E}\left[g^+(X, \beta)\right], \qquad \phi'_-(\beta) = (1-\alpha) - \mathbb{E}\left[g^-(X, \beta)\right].$$

Under the singleton assumption, $g^+(X, \beta) = g^-(X, \beta) = t_\beta(X)$ a.s., hence

$$\phi'_+(\beta) = \phi'_-(\beta) = (1-\alpha) - \mathbb{E}\left[t_\beta(X)\right] = 0.$$

If $\beta > 0$, this implies that $\beta$ is a maximizer of $\phi$. If $\beta = 0$, concavity of $\phi$ and $\phi'_+(0) = 0$ imply $\phi(\beta') \leq \phi(0)$ for all $\beta' \geq 0$, so 0 is also a maximizer. By strong duality (Lemma C.5), $\mathrm{VAL}(\alpha) = \sup_{\beta' \geq 0} \phi(\beta') = \phi(\beta) = \mathbb{E}\big[V_X(t_\beta(X))\big]$, so $t_\beta(\cdot) = g(\cdot, \beta)$ is primal optimal. Therefore, if (13) has multiple solutions, any $\beta^\star \geq 0$ satisfying it yields an optimal $t^\star(x) = g(x, \beta^\star)$.

Finally, if (13) has no solution with $\beta > 0$ but $\mathbb{E}[g(X, 0)] \geq 1 - \alpha$, then necessarily $\mathbb{E}[g(X, 0)] > 1 - \alpha$ (otherwise $\beta = 0$ would solve (13)). In this case there cannot exist a dual maximizer with $\beta^\star > 0$ (since any such maximizer would satisfy (13)), hence any dual maximizer must satisfy $\beta^\star = 0$, and the primal average-coverage constraint is slack. $\square$

**Proof of Proposition 4.1.** Let $Z_i = (X_i, Y_i)$ for $i = 1, \ldots, n$ and $Z_{n+1} = (X_{\text{test}}, Y_{\text{test}})$. For any $\beta \geq 0$ define the empirical coverage fraction

$$\widehat{p}(\beta) := \frac{1}{n+1} \sum_{i=1}^{n+1} \mathbf{1}\{Y_i \in \widehat{C}(X_i; \beta)\}.$$

By definition of Algorithm 1, when we plug in the candidate label $y = Y_{\text{test}}$ we obtain

$$\widehat{\beta}_{Y_{\text{test}}} = \inf\{\beta \geq 0 : \widehat{p}(\beta) \geq 1 - \alpha\},$$

where we select the smallest feasible $\beta$. Since $\widehat{p}(\beta)$ is symmetric in the $n+1$ pairs $\{Z_i\}_{i=1}^{n+1}$, the random variable $\widehat{\beta}_{Y_{\text{test}}}$ is also a symmetric measurable function of $\{Z_i\}_{i=1}^{n+1}$.

Now define $I_i := \mathbf{1}\{Y_i \in \widehat{C}(X_i; \widehat{\beta}_{Y_{\text{test}}})\}$ for $i = 1, \ldots, n+1$. Because the full collection $\{Z_i\}_{i=1}^{n+1}$ is exchangeable and the map from $\{Z_i\}$ to $(I_1, \ldots, I_{n+1})$ is permutation-equivariant, $(I_1, \ldots, I_{n+1})$ is an exchangeable $\{0, 1\}^{n+1}$-valued vector. Therefore, $\mathbb{E}[I_{n+1}] = \mathbb{E}[(n+1)^{-1} \sum_{i=1}^{n+1} I_i]$. Finally, by the defining constraint for $\widehat{\beta}_{Y_{\text{test}}}$ we have $\widehat{p}(\widehat{\beta}_{Y_{\text{test}}}) \geq 1 - \alpha$, i.e. $(n+1)^{-1} \sum_{i=1}^{n+1} I_i \geq 1 - \alpha$, hence

$$\Pr\{Y_{\text{test}} \in C_{\text{ROCP}}(X_{\text{test}})\} = \mathbb{E}[I_{n+1}] = \mathbb{E}\left[\frac{1}{n+1} \sum_{i=1}^{n+1} I_i\right] \geq 1 - \alpha.$$

$\square$

### C.1. Proofs of Auxiliary Lemmas

**Lemma C.4** (Interchange of infimum and expectation for the Lagrangian). *Fix $\beta \geq 0$. Assume $(x, t) \mapsto V_x(t)$ is a normal integrand. Then*

$$\inf_{\substack{t: \mathcal{X} \to [0,1] \\ t \text{ measurable}}} \mathbb{E}\big[V_X(t(X)) - \beta\, t(X)\big] = \mathbb{E}\left[\inf_{u \in [0,1]} \{V_X(u) - \beta u\}\right]. \tag{18}$$

*Moreover, there exists a measurable selector $t_\beta : \mathcal{X} \to [0, 1]$ such that*

$$t_\beta(x) \in \arg\min_{u \in [0,1]} \{V_x(u) - \beta u\} \quad \text{for } P_X\text{-a.e. } x,$$

*and* (18) *holds with $t = t_\beta$.*

**Proof of Lemma C.4.** We first note that all expectations in (18) are well-defined and finite. Fix any $a_0 \in \mathcal{A}$. For $t \in (0,1]$ we have $0 \leq Q_t^x(a_0) \leq M(a_0)$, hence $V_x(t) \leq t Q_t^x(a_0) + (1-t)M(a_0) \leq M(a_0)$. At $t = 0$ we have $V_x(0) = \min_{a \in \mathcal{A}} M(a) \leq M(a_0)$ by definition. Therefore $0 \leq V_x(t) \leq M(a_0)$ for all $(x,t) \in \mathcal{X} \times [0,1]$.

Hence $|V_X(t(X)) - \beta t(X)| \leq M(a_0) + \beta \in L^1(P_X)$. Now, fix $\beta \geq 0$ and define

$$f(x,u) := V_x(u) - \beta u, \qquad (x,u) \in \mathcal{X} \times [0,1], \qquad \underline{f}(x) := \inf_{u \in [0,1]} f(x,u).$$

Since $u \mapsto -\beta u$ is continuous, $f$ is a normal integrand whenever $V$ is. Hence the marginal function $\underline{f}$ is $\mathcal{F}$-measurable (see, e.g., (Rockafellar & Wets, 1998), Thm. 14.37). For any measurable $t : \mathcal{X} \to [0,1]$ we have pointwise $f(x,t(x)) \geq \inf_{u \in [0,1]} f(x,u) = \underline{f}(x)$, and therefore

$$\mathbb{E}\big[f\big(X,t(X)\big)\big] \geq \mathbb{E}\big[\underline{f}(X)\big].$$

Taking the infimum over all measurable $t$ yields the "$\geq$" direction in (18).

For the reverse direction, note that for each $x$ the function $u \mapsto f(x,u)$ has closed epigraph, hence is l.s.c. on the compact interval $[0,1]$ and attains its minimum. Let

$$\Gamma(x) := \arg \min_{u \in [0,1]} f(x,u).$$

Then $\Gamma(x)$ is nonempty and compact. Moreover, because $f$ is a normal integrand, the argmin multifunction $\Gamma : \mathcal{X} \rightrightarrows [0,1]$ has a measurable graph (again see (Rockafellar & Wets, 1998), Thm. 14.37). By the Kuratowski–Ryll–Nardzewski measurable selection theorem (Kuratowski & Ryll-Nardzewski, 1965), there exists a measurable selector $t_\beta : \mathcal{X} \to [0,1]$ such that $t_\beta(x) \in \Gamma(x)$ for $P_X$-a.e. $x$. Consequently, $f\big(x,t_\beta(x)\big) = \underline{f}(x)$ a.e., and integrating yields equality in (18) (and attainment). $\square$

**Lemma C.5** (No duality gap for the average-coverage problem under non-atomic $P_X$)**.** *Assume $P_X$ is non-atomic and that $(x,t) \mapsto V_x(t)$ is a normal integrand. Fixing $\alpha \in (0,1)$, we have*

$$\mathrm{VAL}(\alpha) = \sup_{\beta \geq 0} \phi(\beta),$$

*i.e., there is no duality gap between the primal problem and its Lagrange dual.*

**Proof of Lemma C.5.** We introduce a convex relaxation that allows randomization of $t$ conditional on $X$. Let $\mathcal{P}([0,1])$ be the set of Borel probability measures on $[0,1]$, and let $\mathcal{M}$ be the set of (universally) measurable stochastic kernels $x \mapsto \mu_x \in \mathcal{P}([0,1])$. Consider the relaxed value

$$\mathrm{VAL}_{\mathrm{rel}}(\alpha) := \inf_{\mu_{(\cdot)} \in \mathcal{M}} \Big\{ \mathbb{E}\big[ \textstyle\int V_X(u)\, \mu_X(\mathrm{d}u)\big] : \mathbb{E}\big[\textstyle\int u\, \mu_X(\mathrm{d}u)\big] \geq 1 - \alpha \Big\}. \tag{$\mathrm{P}_{\mathrm{rel}}$}$$

The objective and constraint are linear in $\mu_{(\cdot)}$, so ($\mathrm{P}_{\mathrm{rel}}$) is a convex program. Moreover, $\mu_x = \delta_1$ is strictly feasible when $\alpha > 0$, hence Slater's condition holds. By Fenchel–Rockafellar duality for convex integral functionals with a single linear moment constraint in the framework of normal integrands (see, e.g., (Rockafellar & Wets, 1998) Thm. 11.39.), there is no duality gap for ($\mathrm{P}_{\mathrm{rel}}$) and

$$\mathrm{VAL}_{\mathrm{rel}}(\alpha) = \sup_{\beta \geq 0} \inf_{\mu_{(\cdot)} \in \mathcal{M}} \Big\{ \mathbb{E}\Big[ \textstyle\int \big(V_X(u) - \beta u\big)\, \mu_X(\mathrm{d}u)\Big] + \beta(1-\alpha) \Big\}.$$

Fix $\beta \geq 0$ and write $f_\beta(x,u) := V_x(u) - \beta u$. For any kernel $\mu_{(\cdot)}$ we have pointwise $\int f_\beta(x,u)\, \mu_x(\mathrm{d}u) \geq \inf_{u \in [0,1]} f_\beta(x,u)$, hence

$$\inf_{\mu_{(\cdot)} \in \mathcal{M}} \mathbb{E}\Big[ \textstyle\int f_\beta(X,u)\, \mu_X(\mathrm{d}u)\Big] \geq \mathbb{E}\Big[ \inf_{u \in [0,1]} f_\beta(X,u)\Big].$$

Conversely, by Lemma C.4 there exists a measurable selector $t_\beta$ with $t_\beta(x) \in \arg\min_{u \in [0,1]} f_\beta(x,u)$ for $P_X$-a.e. $x$; taking $\mu_x = \delta_{t_\beta(x)}$ yields equality. Therefore,

$$\inf_{\mu_{(\cdot)} \in \mathcal{M}} \Big\{ \mathbb{E}\Big[ \textstyle\int \big(V_X(u) - \beta u\big)\, \mu_X(\mathrm{d}u)\Big] + \beta(1-\alpha) \Big\} = \beta(1-\alpha) + \mathbb{E}\Big[ \inf_{u \in [0,1]} \big\{V_X(u) - \beta u\big\}\Big] = \phi(\beta),$$

and hence $\mathrm{VAL}_{\mathrm{rel}}(\alpha) = \sup_{\beta \geq 0} \phi(\beta)$.

It remains to show that relaxation does not change the value. Any measurable $t$ yields a feasible kernel $\mu_x = \delta_{t(x)}$, so $\mathrm{VAL}_{\mathrm{rel}}(\alpha) \leq \mathrm{VAL}(\alpha)$. For the reverse inequality, fix any feasible kernel $\mu_{(\cdot)}$ in ($\mathsf{P}_{\mathrm{rel}}$) and consider the two bounded measurable functions $h_1(x, u) = 1 - u$ and $h_2(x, u) = V_x(u)$ on $\mathcal{X} \times [0, 1]$ (boundedness holds in our setting, since $V_x(t) \leq M(a_0)$ for any fixed $a_0 \in \mathcal{A}$), hence also are nonnegative normal integrands. Since $P_X$ is non-atomic, the Dvoretzky–Wald–Wolfowitz purification theorem (Dvoretzky et al., 1951; Balder, 1985) implies that there exists a measurable $t : \mathcal{X} \to [0, 1]$ such that

$$\mathbb{E}\big[V_X\big(t(X)\big)\big] \leq \mathbb{E}\Big[\int V_X(u)\, \mu_X(\mathrm{d}u)\Big], \qquad \mathbb{E}[1 - t(X)] \leq \mathbb{E}\Big[\int (1 - u)\, \mu_X(\mathrm{d}u)\Big].$$

The second inequality rewrites as

$$\mathbb{E}\big[t(X)\big] \ \geq \ \mathbb{E}\Big[\int u\, \mu_X(\mathrm{d}u)\Big] \ \geq \ 1 - \alpha,$$

so $t$ is feasible for (9), while the first inequality shows that its objective value is no larger than that of $\mu(\cdot)$. Hence $\mathrm{VAL}(\alpha) \leq \mathrm{VAL}_{\mathrm{rel}}(\alpha)$. Combining the two inequalities yields $\mathrm{VAL}(\alpha) = \mathrm{VAL}_{\mathrm{rel}}(\alpha) = \sup_{\beta \geq 0} \phi(\beta)$. $\qquad\square$

# D. Experimental implementation details

**ROCP calibration in the experiments.** In our experiments, ROCP is not implemented as a computationally expensive full-conformal retraining procedure: the base predictive model is trained once and then frozen. The exact transductive form in Algorithm 1 calibrates a scalar dual parameter $\widehat{\beta}_y$ for each candidate label $y$, which amounts to $|\mathcal{Y}|$ parallelizable one-dimensional searches. For the reported experiments, we use a split-conformal proxy that calibrates a single global $\beta$ offline on the calibration set. At test time, this evaluates the induced set $\widehat{C}(x; \beta)$ without test-point label-wise optimization, and empirically meets the target coverage in our experiments. Since our experiments have finite label spaces, candidate labels can be enumerated; extending the finite-sample conformalization step to continuous $\mathcal{Y}$ would require a different procedure.

## D.1. Autonomous driving: construction details

This appendix provides the construction of the 3-bit hazard label $Y = (Y_a, Y_\ell, Y_r) \in \{0, 1\}^3$ and the black-box probability model $f_x(y)$ used in the autonomous-driving experiment.

**ROI parametrization.** For any bounding box $b = (x^{(1)}, y^{(1)}, x^{(2)}, y^{(2)})$ in an image of width $W$ and height $H$, we define

$$u = \frac{x^{(1)} + x^{(2)}}{2W}, \qquad v_{\mathrm{bot}} = \frac{y^{(2)}}{H}, \qquad \rho = \frac{y^{(2)} - y^{(1)}}{H},$$

where $u$ is the normalized horizontal center, $v_{\mathrm{bot}}$ is the normalized bottom coordinate, and $\rho$ is the normalized box height. Intuitively, $v_{\mathrm{bot}}$ and $\rho$ serve as simple proxies for proximity: objects that are larger and closer to the bottom of the image are treated as closer to the ego vehicle.

We define the following regions:

$$\text{Left: } u \leq \tfrac{1}{3}, \qquad \text{Right: } u \geq \tfrac{2}{3},$$

$$\text{Ahead-close: } \tfrac{1}{3} \leq u \leq \tfrac{2}{3}, \ \ v_{\mathrm{bot}} \geq 0.6, \ \ \rho \geq 0.15,$$

$$\text{Side-close: } v_{\mathrm{bot}} \geq 0.5, \ \ \rho \geq 0.10.$$

**Ground-truth hazard bits from BDD100K annotations.** For each image, we extract the annotated objects (category and 2D bounding box) from the BDD100K JSON labels. We map categories `person` and `rider` to *pedestrian* objects, and `car`, `truck`, `bus`, `train`, `motor`, `bike` to *vehicle* objects. The hazard bits are defined by the existence of at least one qualifying object in the corresponding ROI:

$$Y_a = 1 \iff \exists \text{ (pedestrian or vehicle) box in the Ahead-close region,}$$

$$Y_\ell = 1 \iff \exists \text{ vehicle box with (Left) \& (Side-close),} \qquad Y_r = 1 \iff \exists \text{ vehicle box with (Right) \& (Side-close).}$$

**Black-box probability model** $f_x$**.** To obtain a black-box probability model $f_x$, we run a pretrained YOLO11 detector (Jocher & Qiu, 2024) on each image and extract three scalar scores: $s_a$ is the maximum confidence among detected pedestrian/vehicle boxes that satisfy the ahead-close rule; $s_\ell$ and $s_r$ are the analogous maxima over detected vehicle boxes in the left/right side-close regions. To convert detector scores into calibrated probabilities, we split the data into training/calibration/test, and we fit an isotonic regression map on training subset, $\phi_k(s) \approx \mathbb{P}(Y_k = 1 \mid s_k = s)$ for each bit $k \in \{a, \ell, r\}$. On the calibration and test subset, we set $p_k(x) = \phi_k(s_k(x))$ and define an 8-class distribution by conditional independence,

$$f_x(y) = \prod_{k \in \{a, \ell, r\}} p_k(x)^{y_k} \big(1 - p_k(x)\big)^{1-y_k}, \ y \in \{0, 1\}^3.$$

**Actions and loss.** We consider actions $\mathcal{A} = \{\mathsf{STOP}, \mathsf{LEFT}, \mathsf{RIGHT}, \mathsf{KEEP}\}$. Let $M \gg 1$ denote a catastrophic collision cost and define, for $y = (y_a, y_\ell, y_r) \in \{0, 1\}^3$, $\ell(\mathsf{KEEP}, y) := M \, \mathbf{1}\{y_a = 1\}$,

$$\ell(\mathsf{LEFT}, y) := M \, \mathbf{1}\{y_\ell = 1\} + c_{\text{turn}} + c_{\text{unnec}} \mathbf{1}\{y_a = 0\},$$

$$\ell(\mathsf{RIGHT}, y) := M \, \mathbf{1}\{y_r = 1\} + c_{\text{turn}} + c_{\text{unnec}} \mathbf{1}\{y_a = 0\},$$

$$\ell(\mathsf{STOP}, y) := c_{\text{stop,free}} \mathbf{1}\{y_a = 0\} + c_{\text{stop,block}} \mathbf{1}\{y_a = 1\}.$$

We use $M = 60$, $c_{\text{turn}} = 3$, $c_{\text{unnec}} = 2$, $c_{\text{stop,free}} = 6$, and $c_{\text{stop,block}} = 2$.

