# OpenReview forum: "Optimal Decision-Making Based on Prediction Sets"
_ICML.cc/2026/Conference — ICML 2026 spotlight_

### Official Review · Reviewer_oSVW · 2026-02-16

**Soundness:** 3
**Presentation:** 3
**Significance:** 3
**Originality:** 3
**Overall Recommendation:** 5
**Confidence:** 4

**Summary:**

This paper addresses the problem of how to make optimal decisions using prediction sets with coverage guarantees. The authors propose Risk-Optimal Conformal Prediction (ROCP), a framework that minimizes worst-case expected loss (risk) against distributions consistent with a prediction set's coverage guarantee.
The key theoretical contributions are:
1. Characterization of the minimax optimal policy for fixed prediction sets.
2. Derivation of oracle-optimal prediction sets via Fenchel-Rockafellar duality.
3. A practical algorithm (ROCP) achieving finite-sample marginal coverage.

**Compliance With Llm Reviewing Policy:**

Affirmed.

**Key Questions For Authors:**

1. How does ROCP behave when $\hat{f}_x$ performs poorly?
2. I feel there is a disconnection between Section 4 and Section 1-3. Section 4.2 mentions that the set constructed by Algorithm 1 has finite sample guarantee, but my understanding is that the goal of the paper is not just to construct any prediction set with marginal guarantee, but to construct a set has guaranteed low risk. The authors did not say much about this aspect in Section 4. The authors did acknowledge that "the approximation guarantee to the optimum is, in general, challenging to establish". Maybe it will be helpful to say a few words on why it is challenging, what are the main difficulties.

**Limitations:**

yes

**Strengths And Weaknesses:**

Soundness
Strengths:
The theoretical framework is technically rigorous. Lemma 2.1 provides a clean closed-form characterization of worst-case expected loss that transparently reveals the trade-off between in-set robustness and out-of-set hedging.
The duality approach in Theorem 3.3 is technically sophisticated, appropriately handling the integral functional structure via normal integrands and Fenchel-Rockafellar duality, which differs from the LP-based approach in Kiyani et al. (2025).
The conformalization in Algorithm 1 guarantees finite-sample marginal coverage under exchangeability, regardless of model quality.
Weaknesses:
The paper's "robustness" claims require careful interpretation. While coverage guarantees are distribution-free, the risk estimates depend on plug-in estimators  $\hat{f}_x$. How would the method's performance suffer when this estimate is bad?
Algorithm 1 seems to assume $\mathcal{Y}$ is discrete. What if $\mathcal{Y}$ is continuous?

Presentation
Strengths:
The paper is generally well-written with clear mathematical exposition. The positioning against Kiyani et al. (2025) is clear and honest about the differences between expected loss and quantile objectives.
Weaknesses:
The distinction between "distribution-free coverage" (guaranteed) and "model-dependent risk estimation" (approximate) could be more prominently highlighted to prevent misinterpretation.

Significance
Strengths:
The problem is important and timely: conformal prediction is widely adopted, but how to use prediction sets for downstream decisions remains underexplored.
The paper makes a meaningful conceptual contribution by showing that expected loss minimization (unlike quantile objectives) requires explicit hedging against out-of-set mass, leading to different optimal policies and set constructions.
Weaknesses:
The scope of impact may be narrower than claimed. The method requires: (a) a well-specified loss function with known catastrophic costs, (b) a reasonably calibrated probabilistic model, and (c) discrete or low-dimensional action spaces where optimization is tractable.
The "autonomous driving" application is somewhat overstated. The experiment uses static images with synthetic loss penalties, not actual sequential decision-making. Without a simulator or causal model, claims about "reducing critical mistakes" in dynamic safety-critical systems are not substantiated.

Originality
Strengths:
The combination of conformal prediction with minimax expected loss is novel. The technical approach (Fenchel-Rockafellar duality via randomized kernels) differs from prior work.

---

> ### Author Rebuttal · Authors · 2026-03-30
>
> Thank you very much for your review. We provide a point-by-point response to your comments below.
>
> ## Weaknesses
>
> > **W1**  The paper's "robustness" claims require ... What if $\mathcal{Y}$ is continuous?
>
> We agree that our robustness claims should be interpreted carefully. The finite-sample marginal coverage guarantee is distribution-free, and does not depend on the quality of the plug-in model $\hat f_x$. By contrast, the risk-targeting part of ROCP is model-dependent: the construction uses $\hat f_x$ to approximate the oracle quantities, so poor estimates can lead to suboptimal coverage allocation and weaker decision quality. However, ROCP is designed to be less brittle than the plug-in Bayes rule that directly minimizes the estimated conditional expected loss under $\hat f_x$. ROCP minimizes a worst-case expected loss over distributions consistent with the coverage constraint, and therefore explicitly accounts for out-of-set mass. When $\hat f_x$ is inaccurate, this protection may still help avoid overly aggressive decisions that place too much trust in a misspecified model, even though proximity to the oracle may deteriorate. We will clarify this distinction between distribution-free coverage and model-dependent risk estimation in the revision.
>
> Regarding the assumption of a discrete label space $\mathcal{Y}$, the decision-theoretic results in Sections 2–3 are stated for general measurable outcome spaces, while the discrete restriction enters only in Algorithm 1 as currently written, because the transductive conformalization step iterates over candidate labels $y \in \mathcal Y$. We will state this explicitly in Section 4.2.  This also matches the experimental settings in the paper. Extending the finite-sample algorithm to continuous $\mathcal Y$ would require a different conformalization procedure and consideration of computational feasibility, which is an interesting direction but beyond the current scope.
>
> > **W2** The distinction between ... to prevent misinterpretation.
>
> We agree with reviewer on this point, and we will revise the paper to make this distinction more explicit. As clarified above, ROCP has two separate components: (i) a finite-sample, distribution-free marginal coverage guarantee, which follows from the conformal calibration step under exchangeability, and (ii) a plug-in approximation to the oracle risk-optimal construction in Sections 3, which uses $\hat f_x$ to estimate the oracle quantities. The current theory in Section 4 establishes (i), but does not provide a distribution-free low-risk guarantee, so poor estimates can lead to suboptimal coverage allocation and weaker decision quality.
>
> > **W3** The scope of impact may be narrower than claimed ... are not substantiated.
>
>  We agree that the practical scope should be stated more narrowly.  Our method is decision-aware, so a user-specified loss function is part of the problem formulation, and its usefulness depends on whether that loss meaningfully captures downstream costs. Likewise, the current implementation assumes that the optimization over actions is tractable: in our experiments the action space is discrete. We will make these assumptions more explicit in the discussion. At the same time, one nuance we will clarify is that a well-calibrated probabilistic model is not required for the finite-sample conformal validity itself: model quality affects how well the method targets the oracle decision objective, not whether the returned set satisfies marginal coverage.
>
> We agree with the reviewer and will narrow the wording accordingly. The BDD100K experiment is intended as a toy autonomous-driving-like static hazard benchmark, not as an evaluation of closed-loop autonomous driving. Its purpose is to stress-test our algorithm and compare to other baselines. We will revise the paper to avoid language suggesting strong safety claims. The empirical takeaway we intend to support is the narrower one: in this static proxy task, accounting for out-of-set mass improves decision quality when out-of-set errors are costly.
>
> ## Questions
>
> > **Q1** How does ROCP behave when $\hat{f}_x$ performs poorly?
>
> Please refer to our response to Weakness 1.
>
> > **Q2** I feel there is a disconnection ... what are the main difficulties.
>
> We agree that Section 4 should explain more clearly why a formal approximation guarantee is challenging. Deriving a rigorous finite-sample excess-risk bound or an oracle-recovery theorem for the plug-in ROCP estimator requires assumptions beyond exchangeability, for example, accurate estimation of the conditional loss quantiles, stability of the action and coverage argmins, and regularity/margin conditions for the value functional. Under such additional conditions, consistency or asymptotic excess-risk guarantees appear plausible. In contrast, a distribution-free optimality theorem is generally unlikely. We will expand our Discussion section to explicitly frame it as an important future direction in the final version

---

> > ### Author Rebuttal · Reviewer_oSVW · 2026-04-03
> >
> > Thank you for address my concerns. I maintain my accept rating

---

> > > ### Author Response · Authors · 2026-04-03
> > >
> > > Thank you very much for your time and constructive feedback. We are glad that our response addressed your concerns, and we will incorporate all suggested revisions into the final paper. Thanks again for reviewing and helping improve our work!

---

### Official Review · Reviewer_jj5Z · 2026-03-11

**Soundness:** 4
**Presentation:** 3
**Significance:** 3
**Originality:** 3
**Overall Recommendation:** 5
**Confidence:** 4

**Summary:**

This paper studies decision theory in a setting where, instead of taking action based on some random variable $Y$, one must take action based on a confidence set $C(X)$ that contains $Y$ with probability $1-\alpha$ (as produced by e.g., conformal prediction procedures). The analysis centers on a worst-case expected loss that is sensitive to the possibility of miscoverage. Results are given that characterize i) optimal policies under this loss and ii) prediction sets that minimize the risk incurred by the downstream optimal policy. The prediction set result serves as inspiration for Risk-Optimal Conformal Prediction (ROCP), which enjoys standard finite-sample conformal guarantees and empirically does well in minimizing downstream loss.

**Compliance With Llm Reviewing Policy:**

Affirmed.

**Final Justification:**

Final recommendation: Accept (5)

This is a well-executed paper with a meaningfully differentiated viewpoint on a natural statistical problem. My only concern (a lack of discussion of the conditional / marginal gap) was adequately addressed in the rebuttal.

**Key Questions For Authors:**

1. How does the present paper compare with Andrews and Chen (2025)?

2. Can feasible decision-making pipelines (i.e. using prediction sets with marginal guarantees) approximate the performance of infeasible ones in a provable manner? Negative results would also be interesting, as would decision theory for prediction sets that have guarantees “between” marginal and conformal.

3. What are the computational properties of ROCP?

**Limitations:**

Yes

**Strengths And Weaknesses:**

**Strengths**
The worst-case expected loss introduced in this paper is well-motivated and differentiated from existing work on decision theory with confidence sets. All theoretical results appear to be sound, and the main text (including the empirical portions) read clearly. The “reverse” question of how to construct distribution-free prediction sets if one plans to use them for decision-making is interesting, and the authors provide a reasonably compelling treatment – ROCP has no proof of optimality, but produces valid prediction sets and takes clear inspiration from the paper’s decision-theoretic analysis.

**Weaknesses**
1. The authors should discuss Andrews and Chen (2025) (https://arxiv.org/abs/2502.17830), which is prior work on decision-making with confidence sets. It would be particularly interesting to understand the relationship, if any, between their extension to “e-certificates” and the notion of expected loss in this paper.
2. The analysis of both optimal actions and prediction sets is done with respect to conditional coverage. In practice, only prediction sets with marginal guarantees are considered. The authors are forthright about this, but do not include any explicit discussion (theoretical or otherwise) of the implications of the marginal / conditional gap.
3. This is minor, but the computational properties of ROCP are unclear. As presented in Algorithm 1, it appears to be a complex full-conformal variant (with a rather involved optimization problem per candidate label). Is this method computationally expensive?

Lastly, this is not a weakness, but I suspect – along the lines of Point 1 – that if relevant existing work has been missed by either the authors or reviewers, it lies in the econometrics literature. For instance, Andrews and Chen (2025) cite Manski (2021) as inspiration. Investigating the econometrics literature may also be helpful from a technical perspective – there are many works on approximating infeasible conditional policies / constraints with applications to statistics, e.g. the conditional conformal work of Gibbs et al. (2025) is inspired by the “conditional-to-marginal” technique of Andrews and Shi (2013).

**References**

[1] Andrews, I., & Chen, J. (2025). Certified Decisions. arXiv preprint arXiv:2502.17830.

[2] Manski, C.F. (2021), Econometrics for Decision Making: Building Foundations Sketched by Haavelmo and Wald. Econometrica, 89: 2827-2853. https://doi.org/10.3982/ECTA17985

[3] Isaac Gibbs, John J Cherian, Emmanuel J Candès, Conformal prediction with conditional guarantees, Journal of the Royal Statistical Society Series B: Statistical Methodology, Volume 87, Issue 4, September 2025, Pages 1100–1126, https://doi.org/10.1093/jrsssb/qkaf008

[4] Andrews, D.W.K. and Shi, X. (2013), Inference Based on Conditional Moment Inequalities. Econometrica, 81: 609-666. https://doi.org/10.3982/ECTA9370

---

> ### Author Rebuttal · Authors · 2026-03-30
>
> Thank you very much for your review. We provide a point-by-point response to your comments below.
>
> ## Weaknesses
>
> > **W1** The authors should discuss ... in this paper.
>
> We will discuss Andrews and Chen (2025) explicitly in the revision. Both papers connect uncertainty quantification to downstream decisions, but they emphasize different objects and criteria. Andrews and Chen (2025) study P-certified decisions in a parameter inference setting, i.e., recommended actions paired with high-probability upper bounds on loss, and show that under their dominance criterion, it is without loss to focus on certificates induced by minimax as-if decisions over confidence sets for an unknown parameter. By contrast, our paper studies prediction sets over labels, derives the minimax-optimal action for a fixed prediction set, characterizes the oracle set-design problem for a worst-case expected-loss objective, and then proposes ROCP as a conformal algorithm targeting those sets. Put differently, their paper centers on certified decisions and the downstream adoption problem, whereas ours centers on the interface between prediction sets and actions together with decision-optimal set construction. The closest conceptual overlap is the minimax-over-set idea, whereas our robust expected-loss objective explicitly accounts for the unconstrained $\alpha$ mass outside the set. Regarding e-certificates, they provide expected-loss control for a recommended action via e-values, whereas in our paper expected loss is the criterion used to derive the action and the prediction set. Connecting the two perspectives could be an interesting direction for future work.
>
> > **W2** The analysis of  ...  the marginal / conditional gap.
>
> Section 2 uses an idealized conditional-coverage oracle only because it yields a clean pointwise game and the closed-form local robust loss in Lemma 2.1/Theorem 2.2. If, instead, one directly replaces the ambiguity class in (1) by the marginal class
> $
> \\mathcal{P}_\\alpha^{\\mathrm{m}}(C)
> :=
> \\left\\{
> P \\text{ on } \\mathcal X \\times \\mathcal Y :
> \\Pr(Y \\in C(X)) \\ge 1-\\alpha
> \\right\\},
> $
> then, for any fixed policy $\\pi$,  similar arguments to those in the proof of Lemma 2.1 yield
> $$ \\sup\_{P \\in \\mathcal{P}\_{\\alpha}^{\\mathrm{m}}(C)} \\mathbb{E}[\\ell(\\pi(C(X)),Y)] = \\bar{\\ell}\_{\\pi}^{\\mathrm{in}} + \\alpha (\\bar{\\ell}\_{\\pi}^{\\mathrm{out}} - \\bar{\\ell}\_{\\pi}^{\\mathrm{in}})\_+ $$
> where $\\bar{\\ell}\_{\\pi}^{\\mathrm{in}} := \\sup\_x \\ell\_{C(x)}^{\\mathrm{in}}(\\pi(C(x)))$ and $\\bar{\\ell}\_{\\pi}^{\\mathrm{out}} := \\sup\_x \\ell\_{C(x)}^{\\mathrm{out}}(\\pi(C(x)))$.
>
> Thus, under marginal validity, the exact robust action game is global: it couples all covariates through the worst in-set and out-of-set losses, and it no longer decomposes pointwise in $x$. In particular,  a strict gap from the local robust loss $L_{C(x)}(a;\\alpha)$ can occur when the worst in-set and worst out-of-set losses arise at different covariates. Moreover, for any fixed prediction set $C$, let
> $
> R_{\\mathrm{cond}}(\\pi)
> :=
> \\sup_x L_{C(x)}(\\pi(C(x));\\alpha)
> $
> denote our original conditional objective, and let $R_{\\mathrm{marg}}(\\pi)$ denote the marginal objective above. Then
>
> $$
> R_{\\mathrm{cond}}(\\pi) \\le R_{\\mathrm{marg}}(\\pi) \\le (2-\\alpha) R_{\\mathrm{cond}}(\\pi).
> $$
>
> Hence, the policy from Theorem 2.2 is a $(2-\\alpha)$-approximation to the optimal marginal policy for the same $C$. By contrast, our set design problem in Section 3 already admits the equivalent marginal coverage reformulation via a coverage assignment $t(x)$ with $\\mathbb{E}[t(X)] \\ge 1-\\alpha$, and ROCP likewise guarantees only finite-sample marginal coverage. We will clarify this marginal/conditional gap in the revision.
>
> > **W3** This is minor ... computationally expensive?
>
> ROCP avoids expensive retraining by training once and then freezing the base predictive model. While Algorithm 1 calibrates a scalar dual parameter $\hat{\beta}_y$ for each candidate label $y$ to establish exact finite-sample marginal coverage, this computation involves only $|\mathcal{Y}|$ parallelizable one-dimensional scalar searches. Furthermore, for practical deployment, we implemented a computationally cheaper split-conformal variant that calibrates a single, global $\beta$ offline solely on the calibration set. At test time, this global-$\beta$ approach evaluates the induced set $\hat{C}(x;\beta)$ without requiring any test-point label-wise optimization (see Supplementary Material for code). Our experiments confirm that this substantially cheaper variant also successfully meets the target coverage. We will explicitly clarify this computational efficiency in the revision.
>
> > **W4** Lastly, this is not a weakness ... Andrews and Shi (2013).
>
> We will add these references in the revision.
>
> ## Questions
>
> **Q1**: Please refer to our response to **W1**
>
> **Q2**: Please refer to our response to **W2**
>
> **Q3**: Please refer to our response to **W3**

---

> > ### Author Rebuttal · Reviewer_jj5Z · 2026-04-03
> >
> > Thank you for the detailed response. I appreciated the discussion of the marginal / conditional gap in particular. I have no outstanding questions, and will accordingly raise my score.

---

> > > ### Author Response · Authors · 2026-04-03
> > >
> > > Thank you very much for your time and constructive feedback. We are glad that our response, especially the discussion of the marginal/conditional gap, addressed your concerns, and we will incorporate all suggested revisions into the final paper. Thanks again for reviewing and helping improve our work!

---

### Official Review · Reviewer_5BZf · 2026-03-12

**Soundness:** 3
**Presentation:** 3
**Significance:** 3
**Originality:** 4
**Overall Recommendation:** 5
**Confidence:** 4

**Summary:**

This paper presents a decision-theoretic framework in which actions are selected to minimize loss based on a prediction set. The authors first investigate the optimal policy for a fixed, conditionally valid prediction set. From this, they derive the optimal prediction set—the one that minimizes robust risk while satisfying a coverage constraint. Finally, they introduce a practical algorithm and evaluate their approach using real-world datasets.

**Compliance With Llm Reviewing Policy:**

Affirmed.

**Final Justification:**

The paper is solid and the authors adequately answer my questions. I keep my positive score.

**Key Questions For Authors:**

Q1. In the article, f is an estimator of the true conditional distribution p(y | x). In general, such an estimator is difficult to calculate when the dimension of X is large. How did you handle this in the experiments?

Q2. The theoretical results apply when C(x) is a valid conditional set. So what is the advantage of producing C_ROCP (which is only valid) rather than simply producing the initial C in equation (14)? In other words, what is the purpose of Proposition 4.1?

Q3. I do not quite understand what the advantage is of solving equation (1) rather than finding the action that minimises the expected loss (equation in the introduction)?

**Limitations:**

No, the limitations of the method are never mentioned.

**Strengths And Weaknesses:**

Strengths:
--

- This is a solid article that addresses a very interesting question. Furthermore, in my opinion, it is clearly written and very enjoyable to read.

- The code is provided for reproducibility purposes.

Weaknesses:
--

- The article is sometimes dense, and the links between the different sections and optimisation problems can be difficult to understand on first reading.

- All the important theoretical results concern cases where  the actual distribution is known. This limits the practicality of the method.

---

> ### Author Rebuttal · Authors · 2026-03-30
>
> Thank you very much for your review. We provide a point-by-point response to your comments below.
>
> ## Weaknesses
>
> > **W1** The article is sometimes dense ... difficult to understand on first reading.
>
> We agree that the connection between the different sections can be made clearer. The intended logic is: Section 2 fixes a prediction set and derives the minimax-optimal action for that set; Section 3 uses the resulting pointwise robust risk to formulate the oracle set-design problem and characterize the optimal coverage allocation; Section 4 then replaces the population quantities by plug-in estimates and conformally calibrates the resulting family to obtain a practical algorithm with finite-sample validity. We will add a clear roadmap of the framework at the end of the introduction and an explicit transition paragraph at the beginning of Section 3 and Section 4 to guide the reader in the revision.
>
> > **W2** All the important theoretical ... This limits the practicality of the method.
>
> We would like to clarify an important distinction. The fixed-set decision result in Section 2 is distribution-agnostic once a set $C(x)$ and coverage level are given: it solves a minimax problem over all distributions consistent with that coverage guarantee. The place where population oracle enters is the oracle set-design problem in Section 3, where quantities such as $Q_t^x(a)$ and the optimal coverage allocation $t^\star(x)$ depend on the true conditional law. Section 4 is then intended to bridge this gap: we replace these quantities by plug-in estimates from a black-box predictive model and then conformally calibrate the resulting family to achieve finite-sample marginal coverage. We agree that this oracle-practical separation should be stated more explicitly, and we will revise the paper accordingly.
>
> ## Questions
> > **Q1** In the article, $f$ ... How did you handle this in the experiments?
>
> In the experiments, we do not attempt to estimate $p(y \mid x)$ directly from the raw high-dimensional input $x$. Instead, we use a task-specific black-box probabilistic predictor $f_x$, and ROCP only uses this predictor as a plug-in estimate of the oracle quantities. In the medical diagnosis experiment, $f_x$ is produced by an ImageNet-pretrained Inception-v3 model fine-tuned on the training split to form a 4-class distribution. In the simulated ``autonomous-driving" experiment, $f_x$ is constructed from a pretrained YOLO11 detector by extracting three detector scores, calibrating them via isotonic regression, and then forming an 8-class distribution. So, in practice, high dimensionality is handled by the underlying learned predictive model. Importantly, the finite-sample coverage guarantee of ROCP does not rely on $f_x$ being correct: $f_x$ is used for risk targeting, while validity comes from the conformal calibration step.
>
> > **Q2** The theoretical results apply ... what is the purpose of Proposition 4.1?
>
> This is an important question. Equation (14) defines the raw plug-in family $\widehat C_0(x;t)$, and after choosing $\widehat g(x,\beta)$ this yields $\widehat C(x;\beta)$. These plug-in sets are oracle-inspired and are designed to target low robust risk, but by themselves they do not come with a distribution-free finite-sample coverage guarantee, because they depend on estimated quantities $(\widehat Q,\widehat a,\widehat g)$ and on the choice of $\beta$. ROCP adds the conformal calibration step that selects $\widehat\beta_y$ from exchangeable calibration data and outputs $C_{\mathrm{ROCP}}$. Proposition 4.1 is precisely the guarantee that this final set has finite-sample marginal coverage $1-\alpha$. So the role of Proposition 4.1 is to turn the oracle-inspired plug-in family into a practically valid finite-sample method.
>
> > **Q3** I do not quite understand ... (equation in the introduction)?
>
> If the true conditional law $P(Y|X=x)$ were known, standard decision theory dictates that we should indeed simply choose the Bayes action: $\arg\min_a \mathbb{E}[l(a,Y)|X=x]$. However, in modern ML pipelines, the true distribution is generally unknown. Instead, the decision-maker receives a conformal prediction set $C(x)$ and its coverage guarantee. Under that weaker information, minimizing the worst-case expected loss over all distributions consistent with the coverage guarantee is the robust counterpart of the Bayes decision problem. The advantage of solving (1) is therefore robustness: the resulting decision rule depends only on guaranteed set-level information, rather than the entire predictive distribution. This matters especially in high-stakes settings with rare but catastrophic out-of-set events. A purely in-set max-min rule ignores the unconstrained $\alpha$-mass entirely, while a Bayes action based on a misspecified model can be brittle.

---

> > ### Author Rebuttal · Reviewer_5BZf · 2026-04-02
> >
> > Thanks for the answers.

---

> > > ### Author Response · Authors · 2026-04-03
> > >
> > > Thank you very much for your time and constructive feedback. We are glad that our response addressed your concerns, and we will incorporate all suggested revisions into the final paper. Thanks again for reviewing and helping improve our work!

---

### Official Review · Reviewer_AXNN · 2026-03-12

**Soundness:** 4
**Presentation:** 4
**Significance:** 3
**Originality:** 3
**Overall Recommendation:** 5
**Confidence:** 4

**Summary:**

The authors provide a new method for prediction set construction designed to optimize a downstream decision-making objective based on in and out of prediction set risk. The authors give a characterization of the optimal prediction set for this objective given an oracle distribution and provide a method to construct prediction sets without knowledge of the oracle distribution through conformal prediction, calling their method risk-optimal conformal prediction (ROCP). They apply their method on a COVID-19 diagnosis dataset and a synthetic dataset mimicking autonomous driving.

**Compliance With Llm Reviewing Policy:**

Affirmed.

**Key Questions For Authors:**

1. Do you think that ROCP is a strict improvement over RAC? The empirical results suggest that ROCP is equal or better than RAC in all cases.

2. Is proving that the estimated ROCP is optimal as an estimate of the oracle prediction set feasible?

3. How does this paper relate to Andrews and Chen (2025), which also discusses the connection between uncertainty quantification and decision-making?

References

Andrews, Isaiah, and Jiafeng Chen. "Certified decisions." arXiv preprint arXiv:2502.17830 (2025).

**Limitations:**

Yes.

**Strengths And Weaknesses:**

Soundness: The submission is technically sound with well-reasoned and thorough theory. The empirical results are clear and well done. The authors are honest about discussing the limitations of their results.

Presentation: The submission is structured well to highlight its place in the literature and is well-written overall, with a clear narrative between sections.

Significance: The authors show that their method is competitive with state-of-the-art prediction sets in optimizing risk while being robust to catastrophic out-of-set tail events. The optimality theory only extends to the oracle prediction set, while the upside of the ROCP estimate is only shown through empirical results.

Originality: This work combines work on conformal prediction sets with risk-minimizing decision making to optimize a prediction set for a downstream risk objective. The work also uses a weighted risk in and and out of the prediction set to better control for tail events.

---

> ### Author Rebuttal · Authors · 2026-03-30
>
> We sincerely thank the reviewer for proposing these insightful questions. We address each of your points in details below.
>
> ## Questions
>
> > **Q1** Do you think that ROCP is a strict improvement over RAC? The empirical results suggest that ROCP is equal or better than RAC in all cases.
>
> We do not view ROCP as a uniform strict improvement over RAC for every problem instance. The two methods are derived for different objectives: RAC is tailored to a quantile-style criterion, whereas ROCP is derived for worst-case expected loss under a coverage constraint. As showed in Lemma 2.1, our robust loss objective takes the form of $L_{S}(a;\alpha)=l_{S}^{in}(a)+\alpha(l_{S}^{out}(a)-l_{S}^{in}(a))_{+}$. This explicitly balances the worst-case loss inside the prediction set with an $\alpha$-weighted penalty for potential catastrophic losses outside the set. When the out-of-set penalty is negligible, the decision rules for ROCP and RAC can coincide, but they differ significantly when rare out-of-set events carry catastrophic losses. This is also consistent with our experiments: under the baseline medical loss matrix, ROCP is close to RAC and can trade slightly higher realized loss for lower critical-mistake rates. However, in the high-stakes asymmetric loss regime (the $10\times$ penalty matrix), RAC becomes brittle as $\alpha$ grows, whereas ROCP remains stable, achieving substantially lower worst-case risk and nearly eliminating critical mistakes.
>
> > **Q2** Is proving that the estimated ROCP is optimal as an estimate of the oracle prediction set feasible?
>
> Our current theory establishes the oracle prediction-set design problem at the population level and proves finite-sample marginal coverage for ROCP under exchangeability. However, establishing a formal approximation guarantee to the optimum is, in general, challenging. Deriving a rigorous finite-sample excess risk bound or an oracle recovery theorem for the plug-in ROCP estimator requires assumptions beyond exchangeability, for example, accurate estimation of the conditional loss quantiles, stability of the action and coverage argmins, and regularity/margin conditions for the value functional. Under such additional conditions, consistency or asymptotic excess risk guarantees appear plausible. In contrast, a distribution-free optimality theorem is generally unlikely. We will expand our Discussion section to explicitly frame it as a future direction in the revision.
>
> > **Q3** How does this paper relate to Andrews and Chen (2025), which also discusses the connection between uncertainty quantification and decision-making?
>
> We agree that Andrews and Chen (2025) is relevant prior work, and we will cite and discuss it explicitly in the revision. Both papers connect uncertainty quantification to downstream decisions, but they emphasize different objects and criteria. Andrews and Chen (2025) study P-certified decisions in a parameter inference setting, i.e., recommended actions paired with high-probability upper bounds on loss, and show that under their dominance criterion, it is without loss to focus on certificates induced by minimax as-if decisions over confidence sets for an unknown parameter. By contrast, our paper studies prediction sets over labels, derives the minimax-optimal action for a fixed prediction set, characterizes the oracle set-design problem for a worst-case expected-loss objective, and then proposes ROCP as a conformal algorithm targeting those sets. Put differently, their paper centers on certified decisions and the downstream adoption problem, whereas ours centers on the interface between prediction sets and actions together with decision-optimal set construction. The closest conceptual overlap is the minimax-over-set idea, whereas our robust expected-loss objective explicitly accounts for the unconstrained $\alpha$ mass outside the set. As a result, the optimal action in our framework is generally not the pure in-set max-min rule.

---

> > ### Author Rebuttal · Reviewer_AXNN · 2026-04-03
> >
> > I thank the authors for the rebuttal. The proposed additions should help add good context. I will maintain my positive evaluation.

---

> > > ### Author Response · Authors · 2026-04-03
> > >
> > > Thank you very much for your time and constructive feedback. We are glad that our response addressed your concerns, and we will incorporate all suggested revisions into the final paper. Thanks again for reviewing and helping improve our work!

---

### Decision · Program_Chairs · 2026-04-30

**Decision:**

Accept (spotlight)

**Comment:**

This paper analyzes decision-making based on conformal prediction sets.  It extends previous work by Kiyani et al. (2025) to consider losses for decisions that are outside the predictions set.  This is an important paper as it performs the first analysis of prediction sets in the usual decision-theoretic formulation.  Moreover, the paper proposes Risk-Optimal Conformal Prediction (ROCP), a concrete implementation for protecting the sets against worst-case risks.  The reviewers unanimously recommended acceptance and stated that all questions were fully resolved in the authors' rebuttal.